# Contextual Bandits With Cross-Learning

**Santiago Balseiro**
Columbia Business School
srb2155@columbia.edu

**Negin Golrezaei**
MIT Sloan School of Management
golrezae@mit.edu

**Mohammad Mahdian**
Google Research
mahdian@google.com

**Vahab Mirrokni**
Google Research
mirrokni@google.com

**Jon Schneider**
Google Research
jschnei@google.com

## Abstract

In the classical contextual bandits problem, in each round $t$, a learner observes some context $c$, chooses some action $a$ to perform, and receives some reward $r_{a,t}(c)$. We consider the variant of this problem where in addition to receiving the reward $r_{a,t}(c)$, the learner also learns the values of $r_{a,t}(c')$ for all other contexts $c'$; i.e., the rewards that would have been achieved by performing that action under different contexts. This variant arises in several strategic settings, such as learning how to bid in non-truthful repeated auctions, which has gained a lot of attention lately as many platforms have switched to running first-price auctions. We call this problem the contextual bandits problem with cross-learning. The best algorithms for the classical contextual bandits problem achieve $\tilde{O}(\sqrt{CKT})$ regret against all stationary policies, where $C$ is the number of contexts, $K$ the number of actions, and $T$ the number of rounds. We demonstrate algorithms for the contextual bandits problem with cross-learning that remove the dependence on $C$ and achieve regret $\tilde{O}(\sqrt{KT})$. We simulate our algorithms on real auction data from an ad exchange running first-price auctions (showing that they outperform traditional contextual bandit algorithms).

## 1 Introduction

In the contextual bandits problem, a learner repeatedly observes some context, takes some action depending on that context, and receives some reward depending on that context. The learner's goal is to maximize their total reward over some number of rounds. The contextual bandits problem is a fundamental problem in online learning: it is a simplified (yet analyzable) variant of reinforcement learning and it captures a large class of repeated decision problems. In addition, the algorithms developed for the contextual bandits problem have been successfully applied in domains like ad placement, news recommendation, and clinical trials [12, 16, 23].

Ideally, one would like an algorithm for the contextual bandits problem which performs approximately as well as the best stationary strategy (i.e., the best fixed mapping from contexts to actions). This can be accomplished by running a separate instance of some low-regret algorithm for the non-contextual bandits problem (e.g. EXP3) for every context. This algorithm achieves regret $\tilde{O}(\sqrt{CKT})$ where $C$ is the number of contexts, $K$ the number of actions, and $T$ the number of rounds. This bound can be shown to be tight [6]. Since the number of contexts can be very large, these algorithms can be impractical to use, and much modern current research on the contextual bandits problem instead aims to achieve low regret with respect to some smaller set of policies [2, 15, 4].

However, some settings possess additional structure between the rewards and contexts which allow one to achieve less than $\tilde{O}(\sqrt{CKT})$ regret while still competing with the best stationary strategy.

In this paper, we look at a specific type of structure we call *cross-learning between contexts* that is particularly common in strategic settings. In variants of the contextual bandits problem with this structure, playing an action $a$ in some context $c$ at round $t$ not only reveals the reward $r_{a,t}(c)$ of playing this action in this context (which the learner receives), but also reveals to the learner the rewards $r_{a,t}(c')$ for every other context $c'$. Some settings where this structure appears include:

- **Bidding in nontruthful auctions:** Consider a bidder trying to learn how to bid in a repeated non-truthful auction (such as a first-price auction). Every round, the bidder receives a (private, independent) value for the current item, and based on this must submit a bid for the item. The auctioneer then collects the bids from all participants, and decides whether to allocate the item to our bidder, and if so, how much to charge the bidder.

  This can be seen as a contextual bandits problem for the bidder where the context $c$ is the bidder's value for the item, the action $a$ is their bid, and their reward is their net utility from the auction: 0 if they do not win, and their value for the item minus their payment $p$ if they do win. Note that this problem also allows for cross-learning between contexts – the net utility $r_{a,t}(c')$ that would have been received if they had value $c'$ instead of value $c$ is just $(c' - p) \cdot \mathbb{1}(\text{win item})$, which is computable from the outcome of the auction.

  The problem of bidding in nontruthful auctions has gained a lot of attention recently as many online advertising platforms have recently switched from running second-price to first-price auctions.[1] In a first-price auction, the highest bidder is the winner and pays their bid (as opposed to second-price auctions where the winner pays the second highest-bid). First-price auctions are nontruthful mechanisms as bidders have incentives to shade bids so that they enjoy a positive utility when they win [22].

- **Multi-armed bandits with exogenous costs:** Consider a multi-armed bandit problem where at the beginning of each round $t$, a cost $s_{i,t}$ of playing arm $i$ at this round is publicly announced. That is, choosing arm $i$ this round results in a net reward of $r_{i,t} - s_{i,t}$. This captures settings where, for example, a buyer must choose every round to buy one of $K$ substitutable goods – he is aware of the price of each good (which might change from round to round) but must learn over time the utility each type of good brings him.

  This is a contextual bandits problem where the context in round $t$ is the $K$ costs $s_{i,t}$ at this time. Cross-learning between contexts is present in this setting: given the net utility of playing action $i$ with a given up-front cost $s_i$, one can infer the net utility of playing $i$ with any other up-front cost $s_i'$.

- **Dynamic pricing with variable cost:** Consider a dynamic pricing problem where a firm offers a service (or sells a product) to a stream of customers who arrive sequentially over time. Consumer have private and independent willingness-to-pay and the cost of serving a customer is exogenously given and customer dependent. After observing the cost, the firm decides on what price to offer to the consumer who decides whether to accept the service at the offered price. The optimal price for each consumer is contingent in the cost; for example, when demand is relatively inelastic consumers that are more costly to serve should be quoted higher prices. This extends dynamic pricing problems to cases where the firm has exogenous costs (see, e.g., [8] for an overview of dynamic pricing problems).

  This is a special case of the multi-armed bandits with exogenous costs problem, and hence an instance of contextual-bandits with cross-learning.

- **Sleeping bandits:** Consider the following variant of "sleeping bandits", where there are $K$ arms and in each round some subset $S_t$ of these arms are awake. The learner can play any arm and observe its reward, but only receives this reward if they play an awake arm. This problem was originally proposed in [13], where one of the motivating applications is ecommerce settings where not all sellers or items (and hence "arms") might be available every round.

  This is a contextual bandits problem where the context is the set $S_t$ of awake arms. Again, cross-learning between contexts is present in this setting: given the observation of the reward of arm $i$, one can infer the received reward for any context $S_t'$ by just checking whether $i \in S_t'$.

- **Repeated Bayesian games with private types:** Consider a player participating in a repeated Bayesian game with private, independent types. Each round the player receives some type for the current game, performs some action, and receives some utility (which depends on their type, their action, and the other players' actions). Again, this can be viewed as a contextual bandit problem where types are contexts, actions are actions, and utilities are rewards, and once again this problem allows for cross-learning between contexts (as long as the player can compute their utility based on their type and all players' actions).

Note that in many of these settings, the number of possible contexts $C$ can be huge: exponential in $K$ or uncountably infinite. This makes the naive $O(\sqrt{CKT})$-regret algorithm undesirable in these settings. We show that in contextual bandits problems with cross-learning, it is possible to design algorithms which completely remove the dependence on the number of contexts $C$ in their regret bound. We consider both settings where the contexts are generated stochastically (from some distribution $\mathcal{D}$ that may or may not be known to the learner) and settings where the contexts are chosen adversarially. Similarly, we also consider settings where the rewards are generated stochastically and settings where they are chosen adversarially. Our results include:

- **Stochastic rewards, stochastic or adversarial contexts**: We design an algorithm called algorithm UCB1.CL with regret of $\tilde{O}(\sqrt{KT})$.
- **Adversarial rewards, stochastic contexts with known distribution**: We design an algorithm called EXP3.CL with regret of $\tilde{O}(\sqrt{KT})$.
- **Adversarial rewards, stochastic contexts with unknown distribution**: We design an algorithm called EXP3.CL-U with regret $\tilde{O}(K^{1/3}T^{2/3})$.
- **Lower bound for adversarial rewards, adversarial contexts**: We show that when both rewards and contexts are controlled by an adversary, any algorithm must obtain regret at least $\tilde{\Omega}(\sqrt{CKT})$.

All of these algorithms are easy to implement, in the sense that they can be obtained via simple modifications from existing multi-armed bandit algorithms like EXP3 and UCB1, and efficient, in the sense that all algorithms run in time at most $O(C + K)$ per round (and for many of the settings mentioned above, this can be further improved to $O(K)$ time per round). Our main technical contribution is our analysis of UCB1.CL, which requires arguing that UCB1 can effectively use the information from cross-learning despite it being drawn from a distribution that differs from the desired exploration distribution. We accomplish this by constructing a linear program whose value upper bounds (one of the terms in) the regret of UCB1.CL, and bounding the value of this linear program.

We then apply our results to some of the applications listed above. In each case, our algorithms obtain optimal regret bounds with asymptotically less regret than a naive application of contextual bandits algorithms. In particular:

- For the problem of learning to bid in a first-price auction, standard contextual bandit algorithms get regret $O(T^{3/4})$. Our algorithms achieve regret $O(T^{2/3})$. This is optimal even when there is only a single context (value).
- For the problem of multi-armed bandits with exogenous costs, standard contextual bandit algorithms get regret $O(T^{(K+1)/(K+2)}K^{1/(K+2)})$. Our algorithms get regret $\tilde{O}(\sqrt{KT})$, which is tight.
- For our variant of sleeping bandits, standard contextual bandit algorithms get regret $\tilde{O}(\sqrt{2^K KT})$. Our algorithms get regret $\tilde{O}(\sqrt{KT})$, which is tight.

Finally, we test the performance of these algorithms on real auction data from a first-price ad exchange, showing that our algorithms outperform traditional bandit algorithms.

## 1.1 Related Work

For a general overview of research on the multi-armed bandit problem, we recommend the reader to the survey by Bubeck and Cesa-Bianchi [6]. Our algorithms build off of pre-existing algorithms in

the bandits literature, such as EXP3 [2] and UCB1 [20, 14]. Contextual bandits were first introduced under that name in [15], although similar ideas were present in previous works (e.g. the EXP4 algorithm was proposed in [2]).

One line of research related to ours studies bandit problems under other structural assumptions on the problem instances which allow for improved regret bounds. Slivkins [21] studies a setting where contexts and actions belong to a joint metric space, and context/action pairs that are close to each other give similar rewards, thus allowing for some amount of "cross-learning". Several works [17, 1] study a partial-feedback variant of the (non-contextual) multi-armed bandit problem where performing some action provides some information on the rewards of performing other actions (thus interpolating between the bandits and experts settings). Our setting can be thought of as a contextual version of this variant. However, since the learner cannot choose the context each round, these two settings are qualitatively different. As far as we are aware, the specific problem of contextual bandits with cross-learning between contexts has not appeared in the literature before.

Recently there has been a surge of interest in applying methods from online learning and bandits to auction design. While the majority of the work in this area has been from the perspective of the auctioneer [19, 18, 7, 9, 11] – learning how to design an auction over time based on bidder behavior – some recent work studies this problem from the perspective of a buyer learning how to bid [24, 10, 5]. In particular, [24] studies the problem of learning to bid in a first-price auction over time, but where the bidder's value remains constant (so there is no context).

## 2 Model and Preliminaries

### 2.1 Multi-armed bandits

In the classic multi-armed bandit problem, a learner chooses one of $K$ arms per round over the course of $T$ rounds. On round $t$, the learner receives some reward $r_{i,t} \in [0, 1]$ for pulling arm $i$ (where the rewards $r_{i,t}$ may be chosen adversarially). The learner's goal is to maximize their total reward.

Let $I_t$ denote the arm pulled by the decision maker at round $t$. The *regret* of an algorithm $A$ for the learner is the random variable $\mathsf{Reg}(A) = \max_i \sum_{t=1}^{T} r_{i,t} - \sum_{t=1}^{T} r_{I_t,t}$. We say an algorithm $A$ for the multi-armed bandit problem is $\delta$-*low-regret* if $\mathbb{E}[\mathsf{Reg}(A)] \leq \delta$ (where the expectation is taken over the randomness of $A$). We say an algorithm $A$ is *low-regret* if it is $\delta$-low-regret for some $\delta = o(T)$. There exist simple multi-armed bandit algorithms which are $\tilde{O}(\sqrt{KT})$-low-regret (e.g. EXP3 when rewards are adversarial, and UCB1 when rewards are stochastic).

### 2.2 Contextual bandits

In our model, we consider a *contextual bandits* problem. In the contextual bandits problem, in each round $t$ the learner is additionally provided with a *context* $c_t$, and the learner now receives reward $r_{i,t}(c)$ if he pulls arm $i$ on round $t$ while having context $c$. The contexts $c_t$ are either chosen adversarially at the beginning of the game or drawn independently each round from some distribution $\mathcal{D}$. Similarly, the rewards $r_{i,t}(c)$ are either chosen adversarially or each independently drawn from some distribution $\mathcal{F}_i(c)$. We assume as is standard that $r_{i,t}(c)$ is always bounded in $[0, 1]$.

In the contextual bandits setting, we now define the regret of an algorithm $A$ in terms of regret against the best stationary policy $\pi$; that is, $\max_{\pi:[C]\to[K]} \sum_{t=1}^{T} r_{\pi(c_t),t}(c_t) - \sum_{t=1}^{T} r_{I_t,t}(c_t)$, where $I_t$ is the arm pulled by $M$ on round $t$. The definition of best stationary policy $\pi$ depends slightly on how contexts and rewards are chosen:

- When rewards are stochastic ($r_{i,t}(c)$ drawn independently from $\mathcal{F}_i(c)$ with mean $\mu_i(c)$), we define $\pi(c) = \arg\max_i \mu_i(c)$.

- When rewards are adversarial but contexts are stochastic, we define $\pi(c)$ to be the stationary policy which maximizes $\mathbb{E}_{c_t \sim \mathcal{D}}[\sum_t r_{\pi(c_t),t}(c_t)]$.

- When both rewards and contexts are adversarial, we define $\pi(c)$ to be the stationary policy which maximizes $\sum_t r_{\pi(c_t),t}(c_t)$.

These choices are unified in the following way: in all of the above cases, $\pi$ is the best stationary policy in expectation for someone who knows all the decisions of the adversary and details of

the system ahead of time, but not the randomness in the instantiations of contexts/rewards from distributions. This matches commonly studied notions of regret in the contextual bandits literature; see Appendix A.1 of the Full Version [3] for further discussion. As before, we say an algorithm is $\delta$-low regret if $\mathbb{E}[\text{Reg}(A)] \leq \delta$, and say an algorithm is low-regret if it is $\delta$-low-regret for some $\delta = o(T)$.

There is a simple way to construct a low-regret algorithm $A'$ for the contextual bandits problem from a low-regret algorithm $A$ for the classic bandits problem: simply maintain a separate instance of $A$ for every different context $c$. In the contextual bandits literature, this is sometimes referred to as the $S$-EXP3 algorithm when $A$ is EXP3 [6]. This algorithm is $\tilde{O}(\sqrt{CKT})$-low-regret. We define the $S$-UCB1 algorithm similarly, which is also $\tilde{O}(\sqrt{CKT})$-low-regret when rewards are generated stochastically.

We consider a variant of the contextual bandits problem we call *contextual bandits with cross-learning*. In this variant, whenever the learner pulls arm $i$ in round $t$ while having context $c$ and receives reward $r_{i,t}(c)$, they also learn the value of $r_{i,t}(c')$ for all other contexts $c'$. We define the notions of regret and low-regret similarly for this problem.

## 3 Cross-learning between contexts

In this section, we present two algorithms for the contextual bandits problem with cross-learning: UCB1.CL, for stochastic rewards and adversarial contexts (Section 3.1), and EXP3.CL, for adversarial rewards and stochastic contexts (Section 3.2). Then, in Section 3.3, we show that it is impossible to achieve regret better than $\tilde{O}(\sqrt{CKT})$ when both rewards and contexts are controlled by an adversary (in particular, when both rewards and contexts are adversarial, cross-learning may not be beneficial at all).

### 3.1 Stochastic rewards

In this section we will present an $O(\sqrt{KT \log K})$ algorithm for the contextual bandits problem with cross learning in the stochastic reward setting: i.e., every reward $r_{i,t}(c)$ is drawn independently from an unknown distribution $\mathcal{F}_i(c)$ supported on $[0, 1]$. Importantly, this algorithm works even when the contexts are chosen adversarially, unlike our algorithms for the adversarial reward setting. We call this algorithm UCB1.CL (Algorithm 1).

---

**Algorithm 1** $O(\sqrt{KT \log K})$ regret algorithm (UCB1.CL) for the contextual bandits problem with cross-learning where rewards are stochastic and contexts are adversarial.

---

1: Define the function $\omega(s) = \sqrt{(2 \log T)/s}$.
2: Pull each arm $i \in [K]$ once (pulling arm $i$ in round $i$).
3: Maintain a counter $\tau_{i,t}$, equal to the number of times arm $i$ has been pulled up to round $t$ (so $\tau_{i,K} = 1$ for all $i$).
4: For all $i \in [K]$ and $c \in [C]$, initialize variable $\sigma_{i,K}(c)$ to $r_{i,i}(c)$. Write $\bar{r}_{i,t}(c)$ as shorthand for $\sigma_{i,t}(c)/\tau_{i,t}$.
5: **for** $t = K + 1$ to $T$ **do**
6:     Receive context $c_t$.
7:     Let $I_t$ be the arm which maximizes $\bar{r}_{I_t,t-1}(c_t) + \omega(\tau_{I_t,t-1})$.
8:     Pull arm $I_t$, receiving reward $r_{I_t,t}(c_t)$, and learning the value of $r_{I_t,t}(c)$ for all $c$.
9:     **for** each $c$ in $[C]$ **do**
10:         Set $\sigma_{I_t,t}(c) = \sigma_{I_t,t-1}(c) + r_{I_t,t}(c)$.
11:     **end for**
12:     Set $\tau_{I_t,t} = \tau_{I_t,t-1} + 1$.
13: **end for**

---

The UCB1.CL algorithm is a straightforward generalization of $S$-UCB1; both algorithms maintain a mean and upper confidence bound for each action in each context, and always choose the action with the highest upper confidence bound (the difference being that UCB1.CL uses cross-learning to update the appropriate means and confidence bounds for all contexts each round). The analysis of UCB1.CL,

however, requires new ideas to deal with the fact that the observations of rewards may be drawn from a very different distribution than the desired exploration distribution.

Very roughly, the analysis is structured as follows. Since rewards are stochastic, in every context $c$, there is a "best arm" $i^*(c)$ that the optimal policy always plays. Every other arm $i$ is some amount $\Delta_i(c)$ worse in expectation than the best arm. After observing this arm $m_i(c) = O(\log(T)/\Delta_i(c)^2)$ times, one can be confident that this arm is not the best arm. We can decompose the regret into the regret incurred "before" and "after" the algorithm is confident that an arm is not optimal in a specific context. The regret "after" can be bounded using standard techniques from the bandit literature. Our main contribution is the bound of the regret "before."

Fix an arm $i$ and let $X_i(c)$ be the number of times the algorithm pulls arm $i$ in context $c$ before pulling arm $i$ a total of $m_i(c)$ times across all contexts. Because once arm $i$ is pulled $m_i(c)$ times we are confident about the optimality of pulling that arm in context $c$, we only need to control the number pulls before $m_i(c)$. Therefore, the regret "before" of arm $i$ is roughly $\sum_c X_i(c)\Delta_i(c)$.

We control the regret "before" by setting up a linear program in the variables $X_i(c)$ with objective $\sum_{c,i} X_i(c)\Delta_i(c)$. Because $X_i(c)$ counts all pulls of arm $i$ before $m_i(c)$ we have that $X_i(c) \leq m_i(c)$. This inequality, while valid, does not lead to a tight bound. To obtain a tighter inequality we first sort the contexts in terms of the samples needed to learn whether an arm is optimal, i.e., in increasing order of $m_i(c)$. Because a different context is realized in every round, we can consider the inequality $\sum_{c':m_i(c') \leq m_i(c)} X_i(c') \leq m_i(c)$, which counts the subset of first $m_i(c)$ pulls of arm $i$. Bounding the value of this objective (by effectively taking the dual), we can show that the total regret is at most $O(\sqrt{T})$.

**Theorem 1** (Regret of UCB1.CL). UCB1.CL *(Algorithm 1) has expected regret* $O(\sqrt{KT\log K})$ *for the contextual bandits problem with cross-learning in the setting with stochastic rewards and adversarial contexts.*

As a consequence of the proof of Theorem 1, we have the following gap-dependent bound on the regret of UCB1.CL.

**Corollary 2** (Gap-dependent Bound for UCB1.CL). *Let* $\Delta_{min} = \min_{i,c} \mu^*(c) - \mu_i(c)$ *(where* $\mu^*(c) = \max_i \mu_i(c)$*). Then* UCB1.CL *(Algorithm 1) has expected regret of* $O\left(\frac{K\log T}{\Delta_{min}}\right)$ *for the contextual bandits problem with cross-learning in the setting with stochastic rewards and adversarial contexts.*

## 3.2 Adversarial rewards and stochastic contexts

We now present a $O(\sqrt{KT\log K})$ regret algorithm for the contextual bandits problem with cross learning when the rewards are adversarially chosen but contexts are stochastically drawn from some distribution $\mathcal{D}$. We call this algorithm EXP3.CL (Algorithm 2). For now we assume the learner knows the distribution over contexts $\mathcal{D}$.

---

**Algorithm 2** $O(\sqrt{KT\log K})$ regret algorithm (EXP3.CL) for the contextual bandits problem with simulated contexts.

---

1: Choose $\alpha = \beta = \sqrt{\frac{\log K}{KT}}$.
2: Initialize $K \cdot C$ weights, one for each pair of action $i$ and context $c$, letting $w_{i,t}(c)$ be the value of the $i$th weight for context $c$ at round $t$. Initially, set all $w_{i,0} = 1$.
3: **for** $t = 1$ to $T$ **do**
4:     Draw context $c_t$ from $\mathcal{D}$.
5:     For all $i \in [K]$ and $c \in [C]$, let $p_{i,t}(c) = (1 - K\alpha)\frac{w_{i,t-1}(c)}{\sum_{j=1}^K w_{j,t-1}(c)} + \alpha$.
6:     Sample an arm $I_t$ from the distribution $p_t(c_t)$.
7:     Pull arm $I_t$, receiving reward $r_{I_t,t}(c_t)$, and learning the value of $r_{I_t,t}(c)$ for all $c$.
8:     **for** each $c$ in $[C]$ **do**
9:         Set $w_{I_t,t}(c) = w_{I_t,t-1}(c) \cdot \exp\left(\beta \cdot \frac{r_{I_t,t}(c)}{\sum_{c'=1}^C \Pr[c'] \cdot p_{I_t,t}(c')}\right)$.
10:     **end for**
11: **end for**

---

Both EXP3.CL and $S$-EXP3 maintain a weight for each action in each context, and update the weights via multiplicative updates by an exponential of an unbiased estimator of the reward. We modify $S$-EXP3 by changing the unbiased estimator in the update rule to take advantage of the information from cross-learning. To minimize regret, we wish to choose an unbiased estimator with minimal variance (as the expected variance of this estimator shows up in the final regret bound). The new estimator in question is

$$\hat{r}_{i,t}(c) = \frac{r_{i,t}(c)}{\sum_{c'=1}^{C} \Pr_{\mathcal{D}}[c'] \cdot p_{i,t}(c')} \cdot \mathbb{1}_{I_t=i}.$$

There are two ways of thinking about this estimator. The first is to note that the denominator of this estimator is exactly the probability of pulling arm $i$ on round $t$ before you learn the realization of $c_t$ (and hence this estimator is unbiased). The second way is to note that for every context $c'$, it is possible to construct an estimator of the form

$$\hat{r}_{i,t}(c) = \frac{r_{i,t}(c)}{\Pr_{\mathcal{D}}[c'] \cdot p_{i,t}(c')} \mathbb{1}_{I_t=i, c_t=c'}.$$

The estimator used in EXP3.CL is the linear combination of these estimators which minimizes variance (i.e. the estimator obtained from importance sampling over this class of estimators). We can show that the total expected variance of this estimator is on the order of $O(\sqrt{KT})$, independent of $C$, implying the following regret bound.

**Theorem 3.** EXP3.CL *(Algorithm 2) has regret $O(\sqrt{TK \log K})$ for the contextual bandits problem with cross learning when rewards are adversarial and contexts are stochastic.*

Calculating this estimator $\hat{r}_{i,t}(c)$ requires the learner to know the distribution $\mathcal{D}$. What can we do if the learner does not know the distribution $\mathcal{D}$? Unlike distributions of rewards (where the learner must actively choose which reward distribution to receive a sample from), the learner receives exactly one sample from $\mathcal{D}$ every round regardless of their action. This suggests the following approach: learn an approximation $\hat{\mathcal{D}}$ to $\mathcal{D}$ by observing the context for some number of rounds, and run EXP3.CL using $\hat{\mathcal{D}}$ to compute estimators. Unfortunately, a straightforward analysis of this approach gives regret that scales as $T^{2/3}$ due to the approximation error in $\hat{\mathcal{D}}$.

In Appendix A.2 of the Full Version [3], we design a learning algorithm EXP3.CL-U which achieves regret $\tilde{O}(K^{1/3}T^{2/3})$ even when the distribution $\mathcal{D}$ is unknown by using a much simpler (but higher variance) estimator that does not require knowledge of $\mathcal{D}$ to compute. It is an interesting open problem whether it is possible to obtain $\tilde{O}(\sqrt{KT})$ regret when $\mathcal{D}$ is unknown.

### 3.3 Adversarial rewards, adversarial contexts

A natural question is whether we can achieve low-regret when both the rewards and contexts are chosen adversarially (but where we still can cross-learn between different contexts). A positive answer to this question would subsume the results of the previous sections. Unfortunately, we will show in this section that any learning algorithm for the contextual bandits problem with cross-learning must necessarily incur $\Omega(\sqrt{CKT})$ regret (which is achieved by $S$-EXP3).

We will need the following regret lower-bound for the (non-contextual) multi-armed bandits problem.

**Lemma 4** (see [2]). *There exists a distribution over instances of the multi-armed bandit problem where any algorithm must incur an expected regret of at least $\Omega(\sqrt{KT})$.*

With this lemma, we can construct the following lower-bound for the contextual bandits problem with cross-learning by connecting $C$ independent copies of these hard instances in sequence with one another so that cross-learning between instances is not possible.

**Theorem 5.** *There exists a distribution over instances of the contextual bandit problem with cross-learning where any algorithm must incur a regret of at least $\Omega(\sqrt{CKT})$.*

# 4 Empirical evaluation

In this section, we empirically evaluate the performance of our contextual bandit algorithms on the problem of learning how to bid in a first-price auction.

Recall that our cross-learning algorithms rely on cross-learning between contexts being possible: if the outcome of the auction remains the same, the bidder can compute their net utility they would receive given any value they could have for the item. This is true if the bidder's value for the item is independent of the other bidders' values for the item. Of course, this assumption (while common in much research in auction theory) does not necessarily hold in practice. We can nonetheless run our contextual bandit algorithms as if this were the case, and compare them to existing contextual bandit algorithms which do not make this assumption.

Our basic experimental setup is as follows. We take existing first-price auction data from a large ad exchange that runs first-price auctions on a significant fraction of traffic, remove one participant (whose true values we have access to), substitute in one of our bandit algorithms for this participant, and replay the auction, hence answering the question "how well would this (now removed) participant do if they instead ran this bandit algorithm?".

We collected anonymized data from 10 million consecutive auctions from this ad exchange, which were then divided into 100 groups of $10^5$ auctions. To remove outliers, bids and values above the 90% quantile were removed, and remaining bids/values were normalized to fit in the $[0, 1]$ interval. We then replayed each group of $10^5$ auctions, comparing the performance of our three algorithms with cross-learning (EXP3.CL-U, EXP3.CL, and UCB1.CL) and the performance of classic contextual bandits algorithms that take no advantage of cross-learning ($S$-EXP3, and $S$-UCB1). Since all algorithms require a discretized set of actions, allowable bids were discretized to multiples of $0.01$. Parameters for each of these algorithms (including level of discretization of contexts for $S$-EXP3 and $S$-UCB1) were optimized via cross-validation on a separate data set of $10^5$ auctions from the same ad exchange.

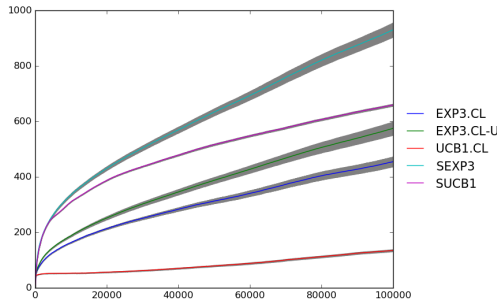

Figure 1: Graph of average cumulative regrets of various learning algorithms (y-axis) versus time (x-axis). Grey regions indicate 95% confidence intervals.

The results of this evaluation are summarized in Figure 1, which plots the average cumulative regret of these algorithms over the $10^5$ rounds. The three algorithms which take advantage of cross-learning (EXP3.CL-U, EXP3.CL, and UCB1.CL) significantly outperform the two algorithms which do not ($S$-EXP3 and $S$-UCB1). Of these, EXP3.CL-U performs the worst, followed by EXP3.CL, followed by UCB1.CL, which vastly outperforms both EXP3.CL-U and EXP3.CL.

What is surprising about these results is that cross-learning works at all, let alone gives an advantage, given that the basic assumption necessary for cross-learning – that your values are independent from other players' bids, so that you can predict what would have happened if your value was different – does not hold. Indeed, for this data, the Pearson correlation coefficient between the values $v$ and the maximum bids $r$ of the other bidders is approximately $0.4$. This suggests that these algorithms are somewhat robust to errors in the cross-learning hypothesis. It is an interesting open question to understand this phenomenon theoretically.

## Footnotes

[1]See  `https://www.blog.google/products/admanager/simplifying-programmatic-first-price-auctions-google-ad-manager/`

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
