[Supplementary Material · Contextual_Bandits_with_Cross_learning__NeurIPS_Full_.pdf]

# Contextual Bandits With Cross-Learning

## Abstract

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

gorithms for partial cross-learning, we can achieve regret $\tilde{O}(\sqrt{KT})$ in the original sleeping bandits setting studied in [16], which recovers their results and is similarly tight.

Finally, we test the performance of these algorithms on real auction data from a first-price ad exchange. In order for cross-learning to be effective in first-price auctions, the bidder should be able to determine the counterfactual utility for different values. That is, after observing the outcome of the auction, the bidder should predict how would their utility change if their value was different. This is possible when the bidder's values are independent of other players' bid. In practice, however, one would expect certain degree of correlation between these quantities and, thus, the independence assumption might not hold. Even though our algorithms do not explicitly account for correlation, numerical results show that our algorithms are somewhat robust to errors in the cross-learning hypothesis and outperform traditional bandit algorithms. We remark that, from the theoretical perspective, when the correlation between values and bids is arbitrary, cross-learning is impossible and the decision maker cannot do better than running a different learning algorithm for each context. A promising research direction is to incorporate correlation by introducing a statistical or behavioral model to capture the dependency between bids and values.

## 1.1 Related Work

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

213 These choices are unified in the following way: in all of the above cases, $\pi$ is the best stationary policy
214 in expectation for someone who knows all the decisions of the adversary and details of the system
215 ahead of time, but not the randomness in the instantiations of contexts/rewards from distributions. This
216 matches commonly studied notions of regret in the contextual bandits literature; see Appendix A.1
217 for further discussion. As before, we say an algorithm is $\delta$-low regret if $\mathbb{E}[\text{Reg}(A)] \leq \delta$, and say an
218 algorithm is low-regret if it is $\delta$-low-regret for some $\delta = o(T)$. The stationary policy in the third
219 definition is sometimes referred as the best policy in hindsight as it considers the best actions that
220 could have been taken after observing all realizations of rewards and contexts. In many applications,
221 however, this benchmark is too strong. Even when contexts are stochastically drawn from a known
222 distribution no algorithm can be shown to achieve sub-linear regret when the number of contexts is
223 large enough (see Theorem 21 in Appendix A.1). Therefore, we adopt the first two benchmarks when
224 contexts are stochastic.

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

*Proof.* We begin by defining the following notation. Let $\mu_i(c)$ be the mean of distribution $\mathcal{F}_i(c)$. Let $i^*(c) = \arg\max_j \mu_j(c)$, and let $\mu^*(c) = \mu_{i^*(c)}(c)$. Let $\Delta_i(c) = \mu^*(c) - \mu_i(c)$ be the gap between the expected reward of playing arm $i$ in context $c$ and of playing the optimal arm $i^*(c)$ in context $c$. As defined in Algorithm 1, let $\tau_{i,t}$ be the number of times arm $i$ has been pulled up to round $t$, and define $\tau_{i,t}(c)$ to be the number of times arm $i$ has been pulled in context $c$ up to round $t$. Note that the regret $\mathrm{Reg}(\mathcal{A})$ of our algorithm is then equal to

$$
\begin{aligned}
\mathrm{Reg}(\mathcal{A}) \;&=\; \sum_{i=1}^{K}\sum_{c=1}^{C} \Delta_i(c)\tau_{i,c}(T) \\
&=\; \sum_{i=1}^{K}\sum_{c=1}^{C}\sum_{t=1}^{T} \Delta_i(c)\mathbb{1}(I_t = i, c_t = c).
\end{aligned}
$$

Define $\Delta_{min} = \sqrt{K\log T/T}$. Note that the sum of all terms in the above expression with $\Delta_i(c) \leq \Delta_{min}$ is at most $\Delta_{min}T$. We can therefore write

$$
\mathrm{Reg}(\mathcal{A}) \leq \Delta_{min}T + \sum_{i=1}^{K}\sum_{c}\sum_{t=1}^{T} \Delta_i(c)\mathbb{1}(I_t = i, c_t = c, \Delta_i(c) \geq \Delta_{min}). \tag{1}
$$

For convenience of notation, from now on, without loss of generality, we assume that all $\Delta_i(c) \geq \Delta_{min}$, and suppress the condition $\Delta_i(c) \geq \Delta_{min}$ in the indicator variables.

Now, define $m_i(c) = \frac{8\log T}{\Delta_i(c)^2}$. This quantity represents the number of times one must pull arm $i$ to observe that $i$ is not the best arm in context $c$ (we will show this later). We thus divide the sum in (1) into two parts. Define:

$$
\mathrm{Reg}_{\mathsf{Pre}} = \sum_{i=1}^{K}\sum_{c=1}^{C}\sum_{t=1}^{T} \Delta_i(c)\mathbb{1}(I_t = i, c_t = c, \tau_{i,t} \leq m_i(c)), \tag{2}
$$

and

$$
\mathrm{Reg}_{\mathsf{Post}} = \sum_{i=1}^{K}\sum_{c=1}^{C}\sum_{t=1}^{T} \Delta_i(c)\mathbb{1}(I_t = i, c_t = c, \tau_{i,t} > m_i(c)). \tag{3}
$$

These two quantities represent the regret incurred before and after (respectively) the algorithm "realizes" an arm is not optimal in a specific context. With these quantities, we can rewrite (1) as

$$
\mathrm{Reg}(\mathcal{A}) \leq \Delta_{min}T + \mathrm{Reg}_{\mathsf{Pre}} + \mathrm{Reg}_{\mathsf{Post}}. \tag{4}
$$

In the following two lemmas, we will now proceed to bound the expected values of $\mathrm{Reg}_{\mathsf{Pre}}$ and $\mathrm{Reg}_{\mathsf{Post}}$. In particular, the following lemma that bounds $\mathbb{E}[\mathrm{Reg}_{\mathsf{Pre}}]$ is our main technical contribution in this proof.

**Lemma 2.** *Let* $\mathrm{Reg}_{\mathsf{Pre}}$ *be the quantity defined in (2). Then,*

$$
\mathbb{E}\left[\mathrm{Reg}_{\mathsf{Pre}}\right] \leq \frac{16K\log T}{\Delta_{min}}.
$$

*Proof.* Fix an action $i$, and order the contexts (that satisfy $\Delta_i(c) \geq \Delta_{min}$) $c_{(1)}, c_{(2)}, \ldots, c_{(n)}$ so that $\Delta_i(c_{(1)}) \geq \Delta_i(c_{(2)}) \geq \cdots \geq \Delta_i(c_{(n)})$. By the definition of $m_i(c)$, this implies that $m_i(c_{(1)}) \leq m_i(c_{(2)}) \leq \cdots \leq m_i(c_{(n)})$. Finally, define

$$
X_i(c) = \sum_{t=1}^{T} \mathbb{1}(c_t = c, I_t = i, \tau_{i,t} \leq m_i(c)).
$$

The quantity $X_i(c)$ can be thought of as the number of times action $i$ is played during context $c$ before the $m_i(c)$th time action $i$ has been played. Note that

$$\sum_{c=1}^{C}\sum_{t=1}^{T}\Delta_i(c)\mathbb{1}(I_t = i, c_t = c, \tau_{i,t} \leq m_i(c)) = \sum_{j=1}^{n}\Delta_i(c_{(j)})X_i(c_{(j)}).$$

On the other hand, by the definition of $X_i(c)$ and the ordering of $m_i(c_{(j)})$, we know that the $X_i(c)$'s satisfy the following system of linear inequalities:

$$X_i(c_{(1)}) \leq m_i(c_{(1)})$$
$$X_i(c_{(1)}) + X_i(c_{(2)}) \leq m_i(c_{(2)})$$
$$\vdots$$
$$X_i(c_{(1)}) + X_i(c_{(2)}) + \cdots + X_i(c_{(n)}) \leq m_i(c_{(n)}). \tag{5}$$

To see why the above inequalities hold, for simplicity, focus on the second inequality (the same argument can be applied for other inequalities). First note that by the fact that $m_i(c_{(1)}) \leq m_i(c_{(2)})$, we have

$$X_i(c_{(1)}) + X_i(c_{(2)}) \leq \sum_{t=1}^{T}\mathbb{1}(c_t = c_{(1)}, I_t = i, \tau_{i,t} \leq m_i(c_{(2)})) + \sum_{t=1}^{T}\mathbb{1}(c_t = c_{(2)}, I_t = i, \tau_{i,t} \leq m_i(c_{(2)}))$$

Further, note that whenever $\mathbb{1}(I_t = i, c_t = c_{(1)}, \tau_{i,t} \leq m) = 1$, then $\mathbb{1}(I_t = i, c_t = c_{(2)}, \tau_{i,t} \leq m) = 0$ and vice versa. This implies that

$$X_i(c_{(1)}) + X_i(c_{(2)}) \leq \sum_{t=1}^{T}\mathbb{1}((c_t = c_{(1)} \text{ or } c_t = c_{(2)}), I_t = i, \tau_{i,t} \leq m_i(c_{(2)})) \leq m_i(c_{(2)}).$$

Now, we wish to bound $\sum_{j}\Delta_i(c_{(j)})X_i(c_{(j)})$. To do this, multiply the $j$th inequality in Eq. (5) through by $\Delta_i(c_{(j)}) - \Delta_i(c_{(j+1)})$ (for the last inequality, just multiply it through by $\Delta_i(c_{(n)})$), and sum all of these inequalities to obtain

$$
\begin{aligned}
\sum_{j=1}^{n}\Delta_i(c_{(j)})X_i(c_{(j)}) &\leq \Delta_i(c_{(n)})m_i(c_{(n)}) + \sum_{j=1}^{n-1}(\Delta_i(c_{(j)}) - \Delta_i(c_{(j+1)}))m_i(c_{(j)}) \\
&= 8\log T\left(\frac{1}{\Delta_i(c_n)} + \sum_{j=1}^{n-1}\frac{\Delta_i(c_{(j)}) - \Delta_i(c_{(j+1)})}{\Delta_i(c_{(j)})^2}\right) \\
&\leq 8\log T\left(\frac{1}{\Delta_{min}} + \int_{\Delta_{min}}^{1}\frac{dx}{x^2}\right) \\
&\leq \frac{16\log T}{\Delta_{min}},
\end{aligned}
$$

where the second equation follows because $m_i(c) = \frac{8\log T}{\Delta_i(c)^2}$, and the third equation holds because $\Delta_i(c_j) \geq \Delta_{min}$ for any $j \in [n]$. Summing this over all $K$ choices of $i$, we obtain our desired bound. $\square$

We next proceed to bound the expected value of $\text{Reg}_{POST}$. This follows from the standard analysis of UCB1.

**Lemma 3.** *Let* $\text{Reg}_{\text{Post}}$ *be the quantity defined in (2). Then,*

$$\mathbb{E}\left[\text{Reg}_{\text{Post}}\right] \leq \frac{K\pi^2}{3}.$$

*Proof.* See appendix. $\square$

Substituting the results of Lemmas 2 and 3 into (11), we obtain

$$\mathbb{E}[\mathsf{Reg}(\mathcal{A})] \leq \Delta_{min}T + \frac{16K\log T}{\Delta_{min}} + \frac{K\pi^2}{3}. \tag{6}$$

Substituting in $\Delta_{min} = \sqrt{K\log T/T}$, it is straightforward to verify that $\mathbb{E}[\mathsf{Reg}(\mathcal{A})] \leq O(\sqrt{KT\log T})$, as desired. $\qquad\square$

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

*Proof.* Divide the $T$ rounds into $C$ epochs of $T/C$ rounds each. Label the $C$ contexts $c_1, c_2, \ldots, c_C$, and adversarially assign contexts so that the context during the $j$th epoch is always $c_j$.

Next, assign rewards so that $r_{i,t}(c) = 0$ if $t$ is in the $j$th epoch and $c \neq c_j$. On the other hand, for $t$ in the $j$th epoch, set rewards $r_{i,t}(c_j)$ according to a hard instance for the multi-armed bandit problem sampled from the distribution from Lemma 6. Call this instance $P_j$, and let $i_j$ be the optimal action to play in $P_j$.

By construction, the best stationary strategy plays $i_j$ whenever the context is $c_j$. In addition, note that cross-learning offers zero additional information here, since all cross-learned rewards will always be 0. Since the hard instances $P_j$ are all independent of each other, any algorithm for the contextual bandits problem with cross-learning which achieves $o(\sqrt{CKT})$ expected regret on this instance must achieve $o(\sqrt{KT/C})$ expected regret on one of the individual instances $P_j$. This contradicts Lemma 6. $\qquad\square$

## 4 Partial cross-learning

So far, we have assumed that our cross-learning between contexts is complete: if we play action $i$ in context $c$, we learn the value of the reward $r_{i,t}(c')$ for all contexts $c'$. In many settings, however, we do not have complete cross-learning, and may only learn the reward $r_{i,t}(c')$ for some subset of contexts $c'$ (e.g. contexts similar to $c$).

In this section we consider the following model of partial cross-learning. For every action $i \in [K]$, we specify a directed graph $G_i$ over the set of contexts $[C]$. An edge $c \to c'$ in $G_i$ indicates that if you play action $i$ in context $c$, you learn the reward of action $i$ in context $c'$. We assume that all self-loops $c \to c$ are present in all graphs $G_i$ (i.e. if you play action $i$ in context $c$ you learn the reward of action $i$ in context $c$).

### 4.1 Graph invariants

Throughout the remainder of this section we will assume that all graphs $G$ are directed and contain all self-loops. Given a vertex $v$ in $G$, let $P(v)$ equal the set of in-neighbors, i.e., vertices $w$ such that there exists an edge $w \to v$, and let $N(v)$ equal the set of vertices of out-neighbors, i.e., $w$ such that there exists an edge $v \to w$ (note that since all our graphs contain self-loops, $v \in N(v)$ and $v \in P(v)$). Before proceeding we define some useful graph-theoretic quantities that will be used to analyze the performace of our algorithms in the partial cross-learning setting.

**Definition 8.** *A* subclique *of a graph $G$ is a subset of vertices $S$ such that for any two vertices $u, v \in S$, there exists an edge $u \to v$. A* clique cover *of a graph $G$ is a partition of its set of vertices into subcliques $S_1, S_2, \ldots, S_r$ (we say $r$ is the size of the clique cover). The* clique covering number $\kappa(G)$ *is the minimum size of a clique cover of $G$.*

**Definition 9.** *An* independent set *in a graph $G$ is a subset of vertices $S$ such that for any two distinct vertices $u, v \in S$, the edge $u \to v$ does not exist in $G$. The* independence number $\iota(G)$ *is the maximum size of an independent set of $G$.*

**Definition 10.** *An* acyclic subgraph *of a graph $G$ is a set of vertices that can be ordered $v_1, v_2, \ldots, v_r$ such that for any $i > j$, there is no edge $v_i \to v_j$. The* maximum acyclic subgraph number $\lambda(G)$ *is the size of the largest acyclic subgraph of $G$.*

**Definition 11.** *The value $\nu_2(G)$ of a graph $G$ (with vertex set $V$) is given by*

$$\nu_2(G) = \sup_{\substack{f:V \to \mathbb{R}^+ \\ \sum f(v)=1}} \left( \sum_{v \in V} \frac{f(v)}{\sqrt{\sum_{w \in P(v)} f(w)}} \right)^2.$$

**Lemma 12.** *For all directed graphs $G$ with self-loops,*

$$\lambda(G) = \sup_{f:V \to \mathbb{R}^+} \sum_{v \in V} \frac{f(v)}{\sum_{w \in P(v)} f(w)}.$$

*Proof.* Denote the right-hand-side of the above expression by $\nu(G)$. We begin by showing that $\nu(G) \geq \lambda(G)$.

Let $(v_1, v_2, \ldots, v_{\lambda(G)})$ be an acyclic subgraph of $G$ of maximum size. Fix a large $M > 1$, and consider the following function $f : V \to \mathbb{R}^+$: $f(v) = M^i$ if $v = v_i$, and $f(v) = 1$ otherwise. We claim that as $M \to \infty$, the quantity

$$\sum_v \frac{f(v)}{\sum_{w \in P(v)} f(w)}$$

approaches a value larger than $\lambda(G)$. To do this, we will simply show that for each $v_i$ in our acyclic subgraph, the quantity

$$\frac{f(v_i)}{\sum_{w \in P(v_i)} f(w)}$$

420    approaches a value larger than $1$.

421    To see this, note that by the definition of an acyclic subgraph, for all $j > i$, there is no edge $v_j \to v_i$.

422    Therefore, for every $w \in P(v_i)$ (with the exception of $P(v_i)$ itself), $f(w) \le M^{i-1}$ because every

423    $w$ in $P(v_i)$ is of the form $v_j$ for some $j < i$, and therefore $\sum_{w \in P(v_i)} f(w) \le |V| M^{i-1} + M^i$. It

424    follows that

$$\frac{f(v_i)}{\sum_{w \in P(v_i)} f(w)} \ge \frac{M^i}{|V| M^{i-1} + M^i}.$$

425    The right hand side of this expression converges to $1$ as $M$ approaches infinity.

426    The proof that $\nu(G) \le \lambda(G)$ follows from Lemma 10 in [2].        $\square$

427    **Lemma 13.** *For all graphs $G$,*

$$\iota(G) \le \nu_2(G) \le \lambda(G) \le \kappa(G).$$

428    *When $G$ is the union of $r$ disjoint cliques, equality holds for all inequalities and all invariants equal $r$.*

429    *Proof.* We prove the inequalities in order.

430    $\underline{\iota(G) \le \nu_2(G)}$:   Let $S$ be an independent set in $G$ of size $\iota(G)$. Define the distribution $f$ via

431    $f(v) = \frac{1-\varepsilon}{\iota(G)}$ (for some small $\varepsilon$) for $v \in S$ and $f(v) = \frac{\varepsilon}{|V| - \iota(G)}$ for $v \notin S$. As $\varepsilon \to 0$, we have that

$$\sum_{v \in V} \frac{f(v)}{\sqrt{\sum_{w \in P(v)} f(w)}} \longrightarrow \sum_{v \in S} \frac{1/\iota(G)}{\sqrt{1/\iota(G)}} = \sqrt{\iota(G)}.$$

432    It follows that

$$\nu_2(G) = \sup \left( \sum_{v \in V} \frac{f(v)}{\sqrt{\sum_{w \in P(v)} f(w)}} \right)^2 \ge \iota(G).$$

433    $\underline{\nu_2(G) \le \lambda(G)}$:   By Cauchy-Schwartz, for any distribution $f$ over $V$, we have that

$$\left( \sum_{v \in V} \frac{f(v)}{\sqrt{\sum_{w \in P(v)} f(w)}} \right)^2 \le \sum_{v \in V} \frac{f(v)}{\sum_{w \in P(v)} f(w)}.$$

434    Taking suprema of both sides, it follows that

$$\nu_2(G) = \sup_f \left( \sum_{v \in V} \frac{f(v)}{\sqrt{\sum_{w \in P(v)} f(w)}} \right)^2 \le \sup_f \sum_{v \in V} \frac{f(v)}{\sum_{w \in P(v)} f(w)} = \lambda(G),$$

435    where the last equality follows from Lemma 12.

436    $\underline{\lambda(G) \le \kappa(G)}$:   Let $(S_1, S_2, \ldots, S_{\kappa(G)})$ be a minimum size clique covering of $G$. Note that no two

437    elements $v, v'$ in the same $S_i$ can belong to the same acyclic subgraph (since by the definition of a

438    clique, there exist edges $v \to v'$ and $v' \to v$). It follows that the size of the largest acyclic subgraph

439    is at most $\kappa(G)$, and thus $\lambda(G) \le \kappa(G)$.

**Unions of cliques** We now show that when $G$ is a disjoint union of $r$ cliques, $\iota(G) = \nu_2(G) = \lambda(G) = \kappa(G) = r$. To do so it suffices (from the above inequalities) to show that $\iota(G) = r$ and $\kappa(G) = r$. The independence number $\iota(G) = r$ since choosing one element from each clique creates an independent set, and any set of $r + 1$ or more vertices must have two vertices from the same clique. The clique covering number $\kappa(G) = r$ since we can cover the graph with the $r$ given cliques, and any covering with fewer than $r$ sets must combine elements in disjoint cliques (thus violating the fact that each set is a clique). $\qquad\square$

## 4.2 Stochastic rewards

In this section we present a low-regret algorithm for the contextual bandits problem with partial crosslearning when rewards are generated stochastically (from some unknown distribution). As with the results in Section 3.1, our low-regret guarantee applies in this case regardless of whether the contexts are generated stochastically or adversarially.

---

**Algorithm 3** $O(\sqrt{\overline{\kappa}KT\log K})$ regret algorithm (UCB1.P-CL) for the contextual bandits problem with partial cross-learning where rewards are stochastic.

---

1: Define the function $\omega(s) = \sqrt{(2\log T)/s}$.
2: Pull each arm $i \in [K]$ once (pulling arm $i$ in turn $i$).
3: Maintain a counter $\tau_{i,t}$, equal to the number of times arm $i$ has been pulled up to round $t$ (so $\tau_{i,K} = 1$ for all $i$).
4: For all $i \in [K]$ and $c \in [C]$, initialize variable $\sigma_{i,K}(c)$ to $r_{i,i}(c)$. Write $\overline{r}_{i,t}(c)$ as shorthand for $\sigma_{i,t}(c)/\tau_{i,t}$.
5: **for** $t = K + 1$ to $T$ **do**
6:     Receive context $c_t$.
7:     Let $I_t$ be the arm which maximizes $\overline{r}_{I_t,t-1}(c_t) + \omega(\tau_{I_t,t-1})$.
8:     Pull arm $I_t$, receiving reward $r_{I_t,t}(c_t)$, and learning the value of $r_{I_t,t}(c)$ for all $c \in N_{I_t}(c_t)$.
9:     **for** each $c$ in $N_{I_t}(c_t)$ **do**
10:         Set $\sigma_{I_t,t}(c) = \sigma_{I_t,t-1}(c) + r_{I_t,t}(c)$.
11:     **end for**
12:     Set $\tau_{I_t,t} = \tau_{I_t,t-1} + 1$.
13: **end for**

---

Like UCB1.CL, our algorithm UCB1.P-CL for the partial cross-learning setting is a straightforward modification of UCB where we simply update all the means that we can every round (that is, we update the means of every outgoing edge in the graph $G_i$). To analyze the regret of this algorithm, let $\overline{\kappa} = \frac{1}{K}\sum_{i\in[K]}\kappa(G_i)$ be the average clique cover size of all graphs $G_i$. We then claim that algorithm UCB1.P-CL incurs at most $O(\sqrt{\overline{\kappa}KT\log K})$ regret.

**Theorem 14.** UCB1.P-CL *(Algorithm 3) has regret* $O(\sqrt{\overline{\kappa}KT\log K})$ *for the contextual bandits problem with partial cross-learning when rewards are stochastic.*

*Proof.* We proceed similarly to the proof of Theorem 1 (and borrow all notation defined in this proof). As before, we have that

$$\mathsf{Reg}(\mathcal{A}) \leq \Delta_{min}T + \sum_{i=1}^{K}\sum_{c=1}^{C}\sum_{t=1}^{T}\Delta_i(c)\mathbb{1}(I_t = i, c_t = c, \Delta_i(c) \geq \Delta_{min}), \qquad (7)$$

and would like to bound the expectation of this latter sum. To do so, we again divide it into two parts ($\mathsf{Reg}_{PRE}$ and $\mathsf{Reg}_{POST}$), but we define these parts differently as in the proof of Theorem 1. Recall that $\tau_{i,t}(c)$ equals the number of times arm $i$ has been pulled in context $c$ up to (and including) round $t$. Define

$$\tau'_{i,t}(c) = \sum_{c'\in P(c)}\tau_{i,t}(c').$$

Note that $\tau'_{i,t}(c)$ is equal to the number of times up to round $t$ we observe the reward of arm $i$ in context $c$. We now define

$$\text{Reg}_{PRE} = \sum_{i=1}^{K} \sum_{c=1}^{C} \sum_{t=1}^{T} \Delta_i(c) \mathbb{1}(I_t = i, c_t = c, \tau'_{i,t}(c) \leq m_i(c)),$$

and

$$\text{Reg}_{POST} = \sum_{i=1}^{K} \sum_{c=1}^{C} \sum_{t=1}^{T} \Delta_i(c) \mathbb{1}(I_t = i, c_t = c, \tau'_{i,t}(c) > m_i(c)).$$

We can then rewrite (7) as

$$\text{Reg}(\mathcal{A}) \leq \Delta_{min} T + \text{Reg}_{PRE} + \text{Reg}_{POST}. \tag{8}$$

We proceed to bound $\mathbb{E}[\text{Reg}_{PRE}]$ and $\mathbb{E}[\text{Reg}_{POST}]$.

**Lemma 15.**

$$\mathbb{E}\left[\text{Reg}_{PRE}\right] \leq \frac{16 \log T \left(\sum_{i=1}^{K} \kappa(G_i)\right)}{\Delta_{min}}.$$

*Proof.* Fix an action $i$, and let $S_1, S_2, \ldots, S_{\kappa(G_i)}$ be a minimal clique covering of the graph $G_i$. Let $r(c)$ equal the value of $r$ such that $c \in S_r$. For each $r \in [\kappa(G_i)]$, define

$$\tilde{\tau}_{i,t}(r) = \sum_{c \in S_r} \tau_{i,t}(c).$$

Note that for all $c$, $S_{r(c)} \subseteq P_i(c)$ (since $S_{r(c)}$ is a clique, all contexts in $S_{r(c)}$ have an edge leading to $c$). It follows that $\tilde{\tau}_{i,t}(r(c)) \leq \tau'_{i,t}(c)$. Now, define $X(c)$ as

$$X(c) = \sum_{t=1}^{T} \mathbb{1}(c_t = c, I_t = i, \tilde{\tau}_{i,t}(r(c)) \leq m_i(c)).$$

Note that since $\tilde{\tau}_{i,t}(r(c)) \leq \tau'_{i,t}(c)$, it is true that

$$\mathbb{1}(c_t = c, I_t = i, \tilde{\tau}_{i,t}(r(c)) \leq m_i(c)) \geq \mathbb{1}(c_t = c, I_t = i, \tau'_{i,t}(c) \leq m_i(c)),$$

and therefore

$$\sum_{c=1}^{C} \sum_{t=1}^{T} \Delta_i(c) \mathbb{1}(I_t = i, c_t = c, \tau'_{i,t}(c) \leq m_i(c)) \leq \sum_{c=1}^{C} \Delta_i(c) X(c).$$

We will now repeat the argument in Lemma 2 (in the analysis of UCB1.CL) for each subclique $S_r$. Fix an $r$, and order the contexts in $S_r$ $c_{(1)}, c_{(2)}, \ldots, c_{(n)}$ so that $\Delta_i(c_{(1)}) \geq \Delta_i(c_{(2)}) \geq \cdots \geq \Delta_i(c_{(n)})$ (and thus $m_i(c_{(1)}) \leq m_i(c_{(2)}) \leq \cdots \leq m_i(c_{(n)})$). From the ordering of the $m_i(c_{(j)})$, we have the following system of inequalities:

$$X(c_{(1)}) \leq m_i(c_{(1)})$$
$$X(c_{(1)}) + X(c_{(2)}) \leq m_i(c_{(2)})$$
$$\vdots$$
$$X(c_{(1)}) + X(c_{(2)}) + \cdots + X(c_{(n)}) \leq m_i(c_{(n)}). \tag{9}$$

Repeating the logic in Lemma 2, these inequalities imply that

$$\sum_{c \in S_r} \Delta_i(c) X(c) \leq \frac{16 \log T}{\Delta_{min}}.$$

Therefore, summing over all $r \in [\kappa(G_i)]$, we have that

$$\sum_c \Delta_i(c) X(c) \leq \frac{16\kappa(G_i) \log T}{\Delta_{min}}.$$

Finally, summing over all actions $i$, we have that

$$\mathsf{Reg}_{PRE} \leq \frac{16 \log T}{\Delta_{min}} \left( \sum_{i=1}^{K} \kappa(G_i) \right).$$

$\square$

We next proceed to bound the expected value of $\mathsf{Reg}_{POST}$. Again, this follows from the standard analysis of UCB1.

**Lemma 16.**

$$\mathbb{E}\left[\mathsf{Reg}_{POST}\right] \leq \frac{K\pi^2}{3}$$

*Proof.* The proof is identical to the proof of Lemma 3. $\square$

Substituting the results of Lemmas 2 and 3 into (8), we obtain

$$\mathbb{E}[\mathsf{Reg}(\mathcal{A})] \leq \Delta_{min} T + \frac{16 K \overline{\kappa} \log T}{\Delta_{min}} + \frac{K\pi^2}{3}.$$

Substituting in $\Delta_{min} = \sqrt{\overline{\kappa} K \log T / T}$, it is straightforward to verify that $\mathbb{E}[\mathsf{Reg}(\mathcal{A})] \leq O(\sqrt{\overline{\kappa} K T \log T})$, as desired. $\square$

### 4.3 Adversarial rewards

---

**Algorithm 4** $O(\sqrt{\overline{\nu} K T \log K})$ regret algorithm (EXP3.P-CL) for the contextual bandits problem with partial cross-learning where rewards are adversarial and contexts are stochastic.

---

1: Choose $\alpha = \beta = \sqrt{\frac{\log K}{\overline{\nu} K T}}$ (where $\overline{\nu} = \frac{1}{K} \sum_{i=1}^{K} \nu(G_i)$).
2: Initialize $K \cdot C$ weights, one for each pair of action $i$ and context $c$, letting $w_{i,t}(c)$ be the value of the $i$th weight for context $c$ at round $t$. Initially, set all $w_{i,0} = 1$.
3: **for** $t = 1$ to $T$ **do**
4:     Draw context $c_t$ from $\mathcal{D}$.
5:     For all $i \in [K]$ and $c \in [C]$, let $p_{i,t}(c) = (1 - K\alpha)\frac{w_{i,t-1}(c)}{\sum_{j=1}^{K} w_{j,t-1}(c)} + \alpha$.
6:     Sample an arm $I_t$ from the distribution $p_t(c_t)$.
7:     Pull arm $I_t$, receiving reward $r_{I_t,t}(c_t)$, and learning the value of $r_{I_t,t}(c)$ for all $c \in N_i(c_t)$.
8:     **for** each $c$ in $N_i(c_t)$ **do**
9:         Set $w_{I_t,t}(c) = w_{I_t,t-1}(c) \cdot \exp\left(\beta \cdot \frac{r_{I_t,t}(c)}{\sum_{c' \in P(c)} \Pr[c'] \cdot p_{I_t,t}(c')}\right)$.
10:    **end for**
11: **end for**

---

In this section we present an algorithm for contextual bandits with partial cross-learning when rewards are adversarial and contexts are stochastic. As with EXP3.CL, this comes down to constructing a low variance unbiased estimator $\hat{r}_{i,t}(c)$ for this setting. Since we no longer learn the reward for all contexts $c$, we cannot use the estimator in EXP3.CL; instead we modify it to the following estimator:

$$\hat{r}_{i,t}(c) = \frac{r_{i,t}(c)}{\sum_{c' \in P_i(c)} \Pr[c'] \cdot p_{i,t}(c')} \mathbb{1}(I_t = i, c_t \in P_i(c)).$$

Let $\bar{\lambda} = \frac{1}{K} \sum_{i \in [K]} \lambda(G_i)$ be the average size of the maximum acyclic subgraph over all graphs $G_i$ (note that since $\lambda(G) \leq \kappa(G)$ for all graphs $G$ by Lemma 13, $\bar{\lambda} \leq \bar{\kappa}$). We will show that EXP3.P-CL obtains at most $O(\sqrt{\bar{\lambda} KT})$ regret.

**Theorem 17.** UCB1.P-CL *(Algorithm 4) has regret* $O(\sqrt{\bar{\lambda} KT \log K})$ *for the contextual bandits problem with partial cross-learning when rewards are stochastic.*

*Proof.* The proof is similar to that of Theorem 23. If we define the estimator

$$\hat{r}_{i,t}(c) = \frac{r_{i,t}(c)}{\sum_{c' \in P_i(c)} \Pr[c'] \cdot p_{i,t}(c')} \mathbb{1}(I_t = i, c_t \in P_i(c)).$$

Note that

$$\Pr[I_t = i, c_t \in P(c)] = \sum_{c' \in P_i(c)} \Pr[c'] \cdot p_{i,t}(c'),$$

so taking expectations over history, we have that

$$\mathbb{E}[\hat{r}_{i,t}(c)] = r_{i,t}(c),$$

and

$$\mathbb{E}[\hat{r}_{i,t}(c)^2] = \frac{r_{i,t}(c)^2}{\sum_{c' \in P_i(c)} \Pr[c'] \cdot p_{i,t}(c')}.$$

Define $W_t(c) = \sum_{i=1}^{K} w_{i,t}(c)$. Now, proceeding in the same way as the proof of Theorem 23, we arrive at the inequalities

$$
\begin{aligned}
\mathbb{E}[\mathsf{Reg}(\mathcal{A})] &\leq \sum_{c=1}^{C} \Pr[c] \left( \frac{\log K}{\beta} + (e-2)\beta \sum_{t=1}^{T} \sum_{i=1}^{K} \mathbb{E}\left[ \frac{p_{i,t}(c)}{\sum_{c' \in P_i(c)} \Pr[c'] \cdot p_{i,t}(c')} \right] r_{i,t}(c)^2 + KT\alpha \right) \\
&= \frac{\log K}{\beta} + (e-2)\beta \sum_{t=1}^{T} \sum_{i=1}^{K} \sum_{c=1}^{C} \Pr[c] \cdot \mathbb{E}\left[ \frac{p_{i,t}(c)}{\sum_{c' \in P_i(c)} \Pr[c'] \cdot p_{i,t}(c')} \right] r_{i,t}(c)^2 + KT\alpha \\
&\leq \frac{\log K}{\beta} + (e-2)\beta \sum_{t=1}^{T} \sum_{i=1}^{K} \mathbb{E}\left[ \sum_{c=1}^{C} \frac{\Pr[c] p_{i,t}(c)}{\sum_{c' \in P_i(c)} \Pr[c'] \cdot p_{i,t}(c')} \right] + KT\alpha \\
&\leq \frac{\log K}{\beta} + (e-2)\beta \sum_{t=1}^{T} \sum_{i=1}^{K} \nu(G_i) + KT\alpha \\
&= \frac{\log K}{\beta} + (e-2)\beta \bar{\nu} KT + KT\alpha \\
&= O(\sqrt{\bar{\nu} KT \log K}).
\end{aligned}
$$

Here we use the fact that $\sum_{c=1}^{C} \frac{\Pr[c] p_{i,t}(c)}{\sum_{c' \in P_i(c)} \Pr[c'] \cdot p_{i,t}(c')} \leq \lambda(G_i)$, since from Lemma 12

$$\lambda(G_i) = \sup_{f:[C] \to \mathbb{R}^+} \sum_{c=1}^{C} \frac{f(c)}{\sum_{c' \in P_i(c)} f(c')},$$

and we can take $f(c) = \Pr[c] \cdot p_{i,t}(c)$. $\qquad \square$

Note that this algorithm requires knowledge of both the distribution over contexts and the feedback graphs $G_i$ over contexts. It is an interesting question whether it is possible to get similar regret bounds when the graphs $G_i$ are unknown.

## 4.4 Lower bounds

In this section we prove some lower bounds on regret for contextual bandits with partial cross-learning that complement the results of the previous two sections. In our lower bounds, we will consider a restricted set of instances where the graph $G_i$ of each arm $i$ is equal to the same graph $G$.

**Theorem 18.** *Any learning algorithm solving the contextual bandits problem with partial cross-learning (for a fixed feedback graph $G$) with stochastic rewards and stochastic contexts must incur expected regret $\Omega(\sqrt{\nu_2(G)KT})$.*

*Proof.* To prove this, we will need a slightly stronger variant of Lemma 6.

**Lemma 19.** *There exists a distribution over instances of the multi-armed bandit problem (with $K$ arms and $T$ rounds) where for any round $t \in [T]$, any algorithm must incur an expected regret of at least $\Omega(\sqrt{K/T})$ in round $t$.*

*Proof.* See Appendix. □

Now, let $f : [C] \to \mathbb{R}^+$ be any distribution on contexts (i.e. $\sum_c f(c) = 1$). Define $g(c) = \sum_{c' \to c} f(c')$. Consider the following distribution over instances of the contextual bandits problem with partial cross-learning:

- Every round, the context $c_t$ is drawn independently from the distribution $f$.

- The distribution of rewards for a context $c$ is drawn from the distribution over hard instances in Lemma 19 for a multi-armed bandit problem with $K$ arms and $g(c)T/2$ rounds.

Note that in the second point, the distribution over reward distributions changes per context depending on $g(c)$. Intuitively, this is because we expect to observe (through cross-learning) the performance of some action in context $c$ in approximately $g(c)T$ rounds.

For each context $c$ and round $t$, let $\tau_c(t) = \sum_{s=1}^{t} \mathbb{1}(c_s \in P(c))$ be the number of rounds up to round $t$ where we observe the performance of some action in context $c$. Let $T_c$ be the total number of rounds $t$ where $c_t = c$ and $\tau_c(t) \le g(c)T$. We claim that we must incur regret at least

$$\Omega\left(\mathbb{E}[T_c]\sqrt{\frac{K}{g(c)T}}\right) \tag{10}$$

from the rounds where $c_t = c$. To see this, let $\{t_1, t_2, \ldots, t_{\min(\tau_c(T), g(c)T)}\}$ be the set of (the first $g(c)T$) rounds where $c_t \to c$, and let $S(c) = \{i|c_{t_i} = c\}$ be the subset of indices where $c_{t_i}$ equals $c$. We claim that, conditioned on $S(c)$, we must incur expected regret at least

$$\Omega\left(|S(c)|\sqrt{\frac{K}{g(c)T}}\right).$$

from the rounds $t_i$ for $i \in S$. If not, this means that there is one $i \in S(c)$ where the expected regret from this round is $o(\sqrt{K/(g(c)T)})$; but this would violate Lemma 19 (in particular, this gives a regular multi-armed bandits algorithm which incurs expected regret $o(\sqrt{K/(g(c)T)})$ in round $i$). Since $|S_c| = T_c$, taking expectations over $T_c$, equation (10) follows.

Now, we claim that $\mathbb{E}[T_c] = \Omega(f(c)T)$. This follows since

$$
\begin{aligned}
\mathbb{E}[T_c] &= \sum_{i=1}^{g(c)T} \Pr[c_{t_i} = c_t] \cdot \Pr[\tau_c(T) \geq i] \\
&= \frac{f(c)}{g(c)} \sum_{i=1}^{g(c)T} \Pr[\tau_c(T) \geq i] \\
&\geq \frac{f(c)}{g(c)} (g(c)T/2) \cdot \Pr[\tau_c(T) \geq g(c)/2] \\
&\geq \frac{f(c)T}{2} \cdot (1 - \exp(-g(c)^2 T/2)) \\
&\geq \Omega(f(c)T)
\end{aligned}
$$

where in the last step, we use that $(1 - \exp(-g(c)^2 T/2)) \geq \Omega(1)$ for sufficiently large $T$.

This implies that the expected regret from rounds where $c_t = c$ is at least $\Omega(f(c)\sqrt{KT/g(c)})$. Summing over all contexts $c$, the total expected regret is at least

$$
\Omega\left( \left( \sum_{c=1}^{C} \frac{f(c)}{\sqrt{g(c)}} \right) \sqrt{KT} \right).
$$

Since $\nu_2(G) = \sup_f \left( \sum_{c=1}^{C} \frac{f(c)}{\sqrt{g(c)}} \right)^2$, taking the supremum over $f$ we find that any algorithm must incur expected regret at least $\Omega(\sqrt{\nu_2(G)KT})$, as desired. $\qquad \square$

When we allow the contexts to be adversarially chosen, we can improve this lower bound to $\Omega(\sqrt{\lambda(G)KT})$.

**Theorem 20.** *Any learning algorithm solving the contextual bandits problem with partial cross-learning (for a fixed feedback graph $G$) with stochastic rewards and adversarial contexts must incur regret $\Omega(\sqrt{\lambda(G)KT})$.*

*Proof.* Let $\{v_1, v_2, \ldots, v_{\lambda(G)}\}$ be a maximum acyclic subset of $G$ (with the property that if $i < j$, there is no edge $v_i \to v_j$). We now proceed as in the proof of Theorem 7. Divide the $T$ rounds into $\lambda(G)$ epochs of $T/\lambda(G)$ rounds each. The adversary must decide both the contexts every round, and the reward distributions for each context. The adversary will do so as follows:

- For each round $t$ in epoch $i$, the adversary will set the context $c_t = v_i$.

- For each context $c$, the adversary will set the reward distribution equal to a hard instance for the multi-armed bandit problem sampled from the distribution from Lemma 6.

Note that since the contexts $v_i$ belong to an acyclic subset of $G$, any information cross-learned in epoch $i$ will reveal nothing about the reward distribution for any context $v_j$ with $j > i$ (and hence nothing about the reward distribution in any epoch $j > i$). Since the hard instances are all independent of each other, any algorithm for the contextual bandits problem with partial cross-learning which achieves $o(\sqrt{\lambda(G)KT})$ expected regret on this instance must achieve $o(\sqrt{KT/\lambda(G)})$ expected regret on one of the individual instances, which contradicts Lemma 6. $\qquad \square$

Note that when the graphs are undirected, $\lambda(G) = \iota(G)$ (since in that case, the definition of acyclic subgraph and independent set coincide), and therefore $\lambda(G) = \nu_2(G) = \iota(G)$ (by Lemma 13). It follows that when all $G_i$ are undirected and equal, the lower bound of Theorem 18 matches the upper bound of Theorem 17 in the setting where contexts are stochastic. Likewise, when $G$ is the disjoint union of $r$ cliques, all of our graph invariants coincide, and our lower bounds are tight. In other settings and for other feedback structures an instance-dependent gap between the best upper bound and best lower bound persists; reducing this gap is an interesting open problem.

## 5  Applications

In this section, we discuss how to apply our results on cross-learning to some of the settings mentioned in the introduction: learning to bid in a first-price auction, multi-armed bandits with exogenous costs, and sleeping bandits. In all cases, we show that the regret bound we obtain by applying the algorithms of Section 3 and Section 4 are optimal (up to $\log T$ factors) and a non-trivial improvement over naively applying $S$-EXP3 or $S$-UCB (possibly discretizing the context space beforehand). We begin by discussing how to efficiently implement our algorithms when the number of contexts is infinite.

### 5.1  Cross-learning between infinitely many contexts

We begin with a brief note on efficiency. Even though the regret bounds we prove in Section 3 do not scale with $C$, note that the computational complexity of all three of our algorithms from Section 3 (EXP3.CL-U, EXP3.CL, and UCB1.CL) scales with the number of contexts $C$: all three algorithms have time complexity $O(C + K)$ per round and space complexity $O(CK)$.

In many of the above settings, the number of contexts can be very large (in some cases, like when the space of contexts is the interval $[0, 1]$, the number of contexts is infinite). However, these settings often also have additional structure which let us run these same algorithms with improved complexity.

Most generally, for all the settings we consider, the observed reward is always an affine function of a straightforward embedding $\rho(c)$ (computable by the learner) of the context into $\mathbb{R}^d$ for some small $d$. That is, for each $i$ and $t$, it is possible to write $r_{i,t}(c) = a_{i,t}^\top \rho(c) + b_{i,t}$, where $a_{i,t} \in \mathbb{R}^d$ and $b_{i,t} \in \mathbb{R}$; moreover, the coefficients $a_{I_t,t}$ and $b_{I_t,t}$ are directly revealed to the learner each round. It in turn follows that the averages $\overline{r}_{i,t}(c)$ stored by UCB1.CL are simply linear functions of $\rho(c)$. Since there is one such function for each arm $i$, this requires a total of $O(Kd)$ space (i.e., we simply store the running averages $\overline{a}_{i,t}$ and $\overline{b}_{i,t}$ and then determine the average reward using the formula $\overline{b}_{i,t} = \overline{a}_{i,t}^\top \rho(c) + \overline{b}_{i,t}$). Similarly, the coefficients can be updated each round in $O(d)$ time simply by updating the average for $I_t$. For example, for $b_{i,t}$ the update is given by

$$\overline{b}_{I_t,t} = \frac{\tau_{I_t,t-1}\overline{b}_{I_t,t-1} + b_{I_t,t}}{\tau_{I_t,t-1} + 1}.$$

Likewise, the weights $w_{i,t}(c)$ stored by EXP3.CL-U, for example, are always of the form $\exp(x_{i,t}\rho(c) + y_{i,t})$, and again it suffices to just maintain a linear function of $\rho(c)$. A similar argument shows that EXP3.CL can be implemented efficiently (with the caveat that to compute the estimators, we must be able to efficiently take expectations over our known distribution on contexts).

### 5.2  Applications of cross-learning

**Bidding in first-price auctions**   In the problem of learning to bid in a first-price auction, every round $t$ (for a total of $T$ rounds) an item is put up for auction. This item has value $v_t \in [0, 1]$ to our bidder. Based on $v_t$, our bidder submits a bid $b_t \in [0, 1]$. Simultaneously, other bidders submit bids for this item; we let $h_t$ be the highest bid of the other bidders in the auction. Finally, if $b_t \geq h_t$, the buyer receives the item and pays $b_t$, obtaining an utility of $v_t - b_t$; otherwise, the buyer does not receive the item and pays nothing, obtaining a utility of zero. The buyer only learns whether or not they receive the item and how much they pay – notably, they do not learn $h_t$ (i.e. this is a non-transparent first price auction). The bidder's goal is to maximize their total utility (total value of items received minus total payment) over the course of $T$ rounds. We assume $v_t$ and $h_t$ are independently drawn each round from distributions $\mathcal{D}_v$ and $\mathcal{D}_h$ respectively, where both distributions are unknown to the bidder.

This can be thought of as a contextual bandits problem, where the contexts are values, the actions are bids, and the rewards are net utilities. Naively applying $S$-UCB to our problem by discretizing the value space and bid space into $C$ and $K$ pieces respectively results in a regret bound of $\tilde{O}(\sqrt{CKT} + T/C + T/K)$ (here the last two terms come from discretization error). Optimizing $C$ and $K$, we find that when $C = K = T^{1/4}$, we can achieve $\tilde{O}(T^{3/4})$ regret in this way.

On the other hand, cross-learning between contexts is possible here (the reward $r_{b_t,t}(v)$ is a known linear function of the value/context $v$), so we can apply UCB1.CL. Doing this (after discretizing the

bid space into $K$ pieces) results in a regret bound of $\tilde{O}(\sqrt{KT} + T/K)$, and optimizing this results in an algorithm which achieves $\tilde{O}(T^{2/3})$ regret. It follows from a reduction to known results about dynamic pricing that any algorithm must incur $\Omega(T^{2/3})$ regret when learning to bid (even when the value $v$ is fixed) – see Appendix A.3 for details.

In the case of bidding in first-price auctions, the decision maker could potentially cross learn across auctions. For example, if the decision maker wins when submitting a bid $b_t$, then a higher bid $b'$ would also win the auction and lead to an utility $v_t - b'$. Conversely, if the decision maker does not win, lower bids would necessarily lose in the auction too. While our algorithm does not explicitly take into account cross-learning across actions, the previous lower bound shows that, in the worst case, cross-learning across actions does not lead to lower regret. An interesting research direction is to design algorithms that exploit both cross-learning across actions and values when the problem has special structure that allows for cross-learning (e.g., the distribution of bids being nicely behaved).

Finally, we emphasize that our algorithms apply when the auctioneer runs other non-truthful auctions.

**Multi-armed bandits with exogenous costs**  In this problem, as in the standard stochastic multi-armed bandit problem, a learner must repeatedly (for $T$ rounds) make a choice between $K$ options, where the reward $r_{i,t} \in [0,1]$ from choosing option $i$ is drawn from some distribution $\mathcal{D}_i$ with mean $\mu_i$. However, in addition to this, at the beginning of each round $t$, a cost $s_{i,t} \in [0,1]$ of playing arm $i$ this round is adversarially chosen and publicly announced (and choosing arm $i$ this round results in a net reward of $r_{i,t} - s_{i,t}$). The learner's goal is to get low regret compared to the optimal strategy, which always chooses the option which maximizes $\mu_i - d_{i,t}$.

This can be thought of as a contextual bandits problem where the context $c_t$ is the cost vector $s_t$. Discretizing the context space $[0,1]^K$ into $(1/\varepsilon)^K$ pieces and running $S$-UCB results in an overall regret bound of $\tilde{O}(\sqrt{TK\varepsilon^{-K}} + \varepsilon T)$. Optimizing this over $\varepsilon$, when $\varepsilon = (K/T)^{1/(K+2)}$, this results in a regret of $\tilde{O}(T^{(K+1)/(K+2)}K^{1/(K+2)})$.

Again, cross-learning between contexts is possible. Applying UCB1.CL, this immediately leads to an algorithm which achieves regret $\tilde{O}(\sqrt{KT})$ (which is optimal since the standard stochastic multi-armed bandit problem is a special case of this problem).

**Sleeping bandits**  In this variant of sleeping bandits, there are $K$ arms (with stochastically generated rewards in $[0,1]$) and in each round some nonempty subset $S_t$ of these arms are awake. The learner can play any arm and observe its reward, but only receives this reward if they play an awake arm. The learner would like to get low regret compared to the best policy (which always plays the awake arm whose distribution has the highest mean).

This is a contextual bandits problem where the context $c_t$ is the set $S_t$ of awake arms. Since there are $2^K - 1$ possible contexts, naively applying $S$-UCB results in an regret bound of $\tilde{O}(\sqrt{2^K KT})$. On the other hand, cross-learning between contexts is again present in this setting: given the observation of the reward of arm $i$, one can infer the received reward for any context $S'_t$ by just checking whether $i \in S'_t$. Applying UCB1.CL, this leads to an optimal $\tilde{O}(\sqrt{KT})$ regret algorithm for this problem.

In the setting of sleeping bandits originally studied by Kleinberg, Niculescu-Mizi, and Sharma, ([16]) the learner can neither play nor observe sleeping arms. We can capture this setting via contextual bandits with partial cross-learning. We adjust the previous setting so that if a learner chooses an asleep arm, they receive zero reward and observe nothing else. Note that in this case, we have the following partial learning structure between contexts:

- If arm $I_t \in S_t$, you learn $r_{I_t,t}(S)$ for all other subsets $S$ (namely, $r_{I_t,t}(S) = \mathbb{1}(I_t \in S)r_{I_t,t}(S_t)$).

- If arm $I_t \notin S_t$, you learn $r_{I_t,t}(S)$ only subsets $S$ where $I_t \notin S$ (where $r_{I_t,t}(S) = 0$).

In other words, $G_i$ is the following graph: there is an edge from $S_1 \to S_2$ if either $i \in S_i$ or if $i \notin S_1 \cup S_2$. Note that $G_i$ has clique cover number $\kappa(G_i) = 2$; the set of subsets containing $i$ and the set of subsets not containing $i$ both form subcliques of $G_i$. It follows from Theorem 14 that running Algorithm 3 results in an optimal regret bound of $\tilde{O}(\sqrt{KT})$.

# 6  Empirical evaluation

In this section, we empirically evaluate the performance of our contextual bandit algorithms on the problem of learning how to bid in a first-price auction.

Recall that our cross-learning algorithms rely on cross-learning between contexts being possible: if the outcome of the auction remains the same, the bidder can compute their net utility they would receive given any value they could have for the item. This is true if the bidder's value for the item is independent of the other bidders' values for the item. Of course, this assumption (while common in much research in auction theory) does not necessarily hold in practice. We can nonetheless run our contextual bandit algorithms as if this were the case, and compare them to existing contextual bandit algorithms which do not make this assumption.

Our basic experimental setup is as follows. We take existing first-price auction data from a large ad exchange that runs first-price auctions on a significant fraction of traffic, remove one participant (whose true values we have access to), substitute in one of our bandit algorithms for this participant, and replay the auction, hence answering the question "how well would this (now removed) participant do if they instead ran this bandit algorithm?".

We collected anonymized data from 10 million consecutive auctions from this ad exchange, which were then divided into 100 groups of $10^5$ auctions. To remove outliers, bids and values above the 90% quantile were removed, and remaining bids/values were normalized to fit in the $[0, 1]$ interval. We then replayed each group of $10^5$ auctions, comparing the performance of our three algorithms with cross-learning (EXP3.CL-U, EXP3.CL, and UCB1.CL) and the performance of classic contextual bandits algorithms that take no advantage of cross-learning ($S$-EXP3, and $S$-UCB1). Since all algorithms require a discretized set of actions, allowable bids were discretized to multiples of $0.01$. Parameters for each of these algorithms (including level of discretization of contexts for $S$-EXP3 and $S$-UCB1) were optimized via cross-validation on a separate data set of $10^5$ auctions from the same ad exchange.

Figure 1: Graph of average cumulative regrets of various learning algorithms (y-axis) versus time (x-axis). Grey regions indicate 95% confidence intervals.

The results of this evaluation are summarized in Figure 1, which plots the average cumulative regret of these algorithms over the $10^5$ rounds. The three algorithms which take advantage of cross-learning (EXP3.CL-U, EXP3.CL, and UCB1.CL) significantly outperform the two algorithms which do not ($S$-EXP3 and $S$-UCB1). Of these, EXP3.CL-U performs the worst, followed by EXP3.CL, followed by UCB1.CL, which vastly outperforms both EXP3.CL-U and EXP3.CL.

What is surprising about these results is that cross-learning works at all, let alone gives an advantage, given that the basic assumption necessary for cross-learning – that your values are independent from other players' bids, so that you can predict what would have happened if your value was different – does not hold. Indeed, for this data, the Pearson correlation coefficient between the values $v$ and the maximum bids $r$ of the other bidders is approximately $0.4$. This suggests that these algorithms are somewhat robust to errors in the cross-learning hypothesis. It is an interesting open question to understand this phenomenon theoretically.

## Footnotes

[1]See `https://www.blog.google/products/admanager/simplifying-programmatic-first-price-auctions-google-ad-manager/`

[2]Unless otherwise specified, all expectations of quantities at time $t$ are taken conditioned on the history of the previous $t - 1$ rounds.

[3]Note that for $T \geq K \log K$, $\alpha \leq 1/K$, so $1 - K\alpha$ is always positive.

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

# A  Appendix

## A.1  Regret in contextual bandits

We define the regret of an algorithm $A$ in the contextual setting as the difference between the performance of our algorithm and the performance of the best stationary strategy $\pi$. In other words,

$$\mathsf{Reg}(A) = \sum_{t=1}^{T} r_{\pi(c_t),t}(c_t) - \sum_{t=1}^{T} r_{I_t,t}(c_t).$$

However, when contexts are stochastic, there are two different natural ways to define "the best stationary strategy" $\pi$. The first maximizes the reward of this strategy for the specific contexts $c_t$ we observed in our run of algorithm $A$:

$$\pi(c) = \arg\max_i \sum_{t=1}^{T} r_{i,t}(c) \mathbb{1}_{c_t=c}$$

The second way simply maximizes the reward of this strategy in expectation over all time:

$$\pi'(c) = \arg\max_i \sum_{t=1}^{T} r_{i,t}(c)$$

These two stationary strategies give rise to two different definitions of regret. We call the regret against strategy $\pi$ the *ex post regret* $\mathsf{Reg}_{post}(A)$ (and denote the associated strategy by $\pi_{post}$), and we call the regret againtst strategy $\pi'$ the *ex ante regret*, $\mathsf{Reg}_{ante}(A)$ (and denote the associated strategy by $\pi_{ante}$). This captures the idea that to the adversary at the beginning of the game (who knows all the rewards, but not when each context will occur), the best stationary strategy in expectation is $\pi_{ante}$. On the other hand, after the game has finished, the best stationary strategy in hindsight is $\pi_{post}$.

In this paper, all bounds we show are for *ex ante regret* (unless otherwise stated, e.g. in Section 3.1). One reason for this is that, while it is possible to eliminate the dependence on $C$ in the ex ante regret,

778 it is impossible to do so for the ex post regret. In particular, for a large enough number of different
779 contexts $C$, it is impossible to get ex post regret that is sublinear in $T$.

780 **Theorem 21.** *For any algorithm $A$, there is an instance of the contextual bandits problem with*
781 *cross-learning where $\mathbb{E}[\mathsf{Reg}_{post}(A)] \geq T/2$.*

782 *Proof.* We will consider an instance of the problem where there are $K = 2$ actions and $C$ contexts,
783 where the distribution $\mathcal{D}$ is uniform over all $C$ contexts. We will choose $C$ to be large enough so that
784 with high probability all the observed contexts $c_t$ are distinct.

785 The adversary will assign rewards as follows. For each round $t$ and context $c$, with probability $1/2$ he
786 will set $r_{1,t}(c) = 1$ and $r_{2,t}(c) = 0$, and with probability $1/2$ he will set $r_{1,t}(c) = 0$ and $r_{2,t}(c) = 1$.

787 Now consider the best strategy $\pi_{post}$ in hindsight. Since each context only appears once, and since
788 there is always an arm with reward 1, for any context and any time, $\pi_{post}$ will receive total reward $T$.
789 On the other hand, since each $r_{i,t}$ is completely independent of the rewards from previous rounds,
790 the maximum expected reward any learning algorithm can guarantee is $T/2$. It follows that $M$ must
791 have $\mathsf{Reg}_{post}(A)$ at least $T/2$. □

792 On the other hand, in many settings, the strategies $\pi_{post}$ and $\pi_{ante}$ agree with high probability,
793 and therefore the two notions of regret $\mathsf{Reg}_{ante}(A)$ and $\mathsf{Reg}_{post}(A)$ are similar in expectation. For
794 example, this occurs when each context occurs often enough.

795 **Theorem 22.** *For each context $c$, let $\Delta_c = \min_{i \neq \pi_{ante}(c)} \frac{1}{T} \sum_t (r_{\pi_{ante}(c),t}(c) - r_{i,t}(c))$, and let*
796 $M = \min_c \Pr[c] \cdot \Delta_c$. If $M \geq \sqrt{2 \log(TCK)/T}$, then $\left| \mathbb{E}[\mathsf{Reg}_{ante}(A)] - \mathbb{E}[\mathsf{Reg}_{post}(A)] \right| \leq 1$.

797 *Proof.* We will show that the probability that $\pi_{ante} \neq \pi_{post}$ is at most $\frac{1}{T}$, from which the result
798 follows.

799 Fix a context $c$, and consider the probability that $\pi_{post}(c) = i \neq \pi_{ante}(c)$. For this to happen, it must
800 be the case that

$$\sum_{t=1}^{T} (r_{\pi_{ante}(c),t}(c) - r_{i,t}(c)) \mathbb{1}_{c_t = c} < 0.$$

801 Since each $\mathbb{1}_{c_t = c}$ is an independent Bernoulli random variable with probability $\Pr[c]$, we have

$$
\begin{aligned}
\mathbb{E}_{\mathcal{D}} \left[ \sum_{t=1}^{T} (r_{\pi_{ante}(c),t}(c) - r_{i,t}(c)) \mathbb{1}_{c_t = c} < 0 \right] &= \Pr[c] \sum_{t=1}^{T} (r_{\pi_{ante}(c),t}(c) - r_{i,t}(c)) \\
&\leq -\Pr[c] \Delta_c \\
&\leq -M,
\end{aligned}
$$

802 It follows from Hoeffding's inequality (and our assumption that $M \geq \sqrt{2T \log(TCK)}$) that

$$\Pr \left[ \sum_{t=1}^{T} (r_{\pi_{ante}(c),t}(c) - r_{i,t}(c)) \mathbb{1}_{c_t = c} < 0 \right] \leq \exp \left( -\frac{A^2}{2T} \right) \leq \frac{1}{TCK}.$$

803 Taking the union bound over all alternate actions $i$ and all possible contexts $c$, we find that $\Pr[\pi_{ante} \neq$
804 $\pi_{post}] \leq \frac{1}{T}$, as desired. □

805 Throughout the entire paper (unless otherwise specified) we work entirely with ex ante regret unless
806 otherwise specified, and suppress subscripts and write $\mathsf{Reg}_{ante}(A)$ as $\mathsf{Reg}(A)$ and $\pi_{ante}(A)$ as $\pi(A)$.

## A.2   EXP3.CL-U: Adversarial rewards, stochastic contexts with unknown distribution

In this section we present an $\tilde{O}(K^{1/3}T^{2/3})$ regret algorithm for the contextual bandits problem with cross-learning in the setting when rewards are adversarial and contexts are stochastic, but when the learner does not know the distribution $\mathcal{D}$ over contexts. We call this algorithm EXP3.CL-U (see Algorithm 5).

---

**Algorithm 5** $\tilde{O}(K^{1/3}T^{2/3})$ regret algorithm (EXP3.CL-U) for the contextual bandits problem with cross-learning when the distribution $\mathcal{D}$ over contexts is unknown.

1: Choose $\alpha = (\log K / K^2 T)^{1/3}$, and $\beta = \sqrt{\frac{\alpha \log K}{T}}$.
2: Initialize $K \cdot C$ weights, one for each pair of action $i$ and context $c$, letting $w_{i,t}(c)$ be the value of the $i$th weight for context $c$ at round $t$. Initially, set all $w_{i,0} = 1$.
3: **for** $t = 1$ to $T$ **do**
4:   Observe context $c_t \sim \mathcal{D}$.
5:   For all $i \in [K]$ and $c \in [C]$, let $p_{i,t}(c) = (1 - K\alpha)\frac{w_{i,t-1}(c)}{\sum_{j=1}^{K} w_{j,t-1}(c)} + \alpha$.
6:   Sample an arm $I_t$ from the distribution $p_t(c_t)$.
7:   Pull arm $I_t$, receiving reward $r_{I_t,t}(c_t)$, and learning the value of $r_{I_t,t}(c)$ for all $c$.
8:   **for** each $c$ in $[C]$ **do**
9:     Set $w_{I_t,t}(c) = w_{I_t,t-1}(c) \cdot \exp\left(\beta \cdot \frac{r_{I_t,t}(c)}{p_{I_t,t}(c_t)}\right)$.
10:   **end for**
11: **end for**

---

EXP3.CL-U is similar to $S$-EXP3, in that both algorithms maintain a weight for each action in each context, and update the weights via multiplicative updates by an exponential of an unbiased estimator of the reward. The main difference between these two algorithms is that while $S$-EXP3 only updates the weight of the chosen action for the current context (i.e. $w_{I_t,t}(c_t)$), EXP3.CL-U uses the information from cross-learning to update the weight of the chosen action for all contexts. More formally, note that for EXP3 $\hat{r}_{i,t}(c) = (r_{i,t}(c)/p_{i,t}(c_t))\mathbb{1}(I_t = i)$ is an unbiased estimator (over the algorithm's randomness) of the reward the adversary chooses from pulling arm $i$ in context $c$, where $p_{i,t}(c)$ is the probability the algorithm chooses action $i$ in round $t$ if the context is $c$. Each round, EXP3.CL-U updates the weight $w_{I_t,t}(c)$ by multiplying it $\exp(\beta \hat{r}_{i,t}(c))$ (whereas $S$-EXP3 does this only for $w_{I_t,t}(c_t)$).

Why does EXP3.CL-U have regret of order $T^{2/3}$ when the dependence on $T$ in $S$-EXP3 is only of order $\sqrt{T}$? The answer lies in understanding how the variance of the unbiased estimator used affects the regret bound of the algorithm. In the analysis of EXP3, one of the quantities in the regret bound is the *total expected variance of the unbiased estimator*. In $S$-EXP3, this quantity takes the form

$$\sum_{t=1}^{T} p_{i,t}(c_t) \mathbb{E}[\hat{r}_{i,t}(c)^2] = \sum_{t=1}^{T} \frac{p_{i,t}(c_t)}{p_{i,t}(c_t)}\hat{r}_{i,t}(c)^2 = \sum_{t=1}^{T} \hat{r}_{i,t}(c)^2 \leq T.$$

However, in EXP3.CL-U (where the desired exploration distribution $p_{i,t}(c)$ can differ from the exploration distribution due to cross-learning), this quantity becomes

$$\sum_{t=1}^{T} p_{i,t}(c) \mathbb{E}[\hat{r}_{i,t}(c)^2] = \sum_{t=1}^{T} \frac{p_{i,t}(c)}{p_{i,t}(c_t)}\hat{r}_{i,t}(c)^2 \leq \frac{T}{\min p_{i,t}(c)}.$$

Optimizing $\min p_{i,t}(c)$ (through selecting the parameter $\alpha$) leads to an $\tilde{O}(T^{2/3}K^{1/3})$ regret bound.

**Theorem 23.** EXP3.CL-U *(Algorithm 5) has regret* $O(K^{1/3}T^{2/3}(\log K)^{1/3})$ *for the contextual bandits problem with cross-learning.*

*Proof.* We proceed similarly to the analysis of EXP3. Begin by defining

$$\hat{r}_{i,t}(c) = \frac{r_{i,t}(c)}{p_{i,t}(c_t)} \mathbb{1}(I_t = i).$$

Note that since $\Pr[I_t = i | c_t = c] = p_{i,t}(c_t)$, the expectation[2] $\mathbb{E}[\hat{r}_{i,t}(c)] = r_{i,t}(c)$ and thus $\hat{r}_{i,t}(c)$ is an unbiased estimator of $r_{i,t}(c)$. In addition, since $p_{i,t}(c) \geq \alpha$, we can bound the variance of $\hat{r}_{i,t}(c)$ via

$$\mathbb{E}\left[\hat{r}_{i,t}(c)^2\right] = \frac{r_{i,t}(c)^2}{p_{i,t}(c_t)} \leq \frac{r_{i,t}(c)^2}{\alpha}. \tag{11}$$

Now, let $W_t(c) = \sum_{i=1}^{K} w_{i,t}(c)$. Note that

$$
\begin{aligned}
\frac{W_{t+1}(c)}{W_t(c)} &= \sum_{i=1}^{K} \frac{w_{i,t}(c)}{W_t(c)} \cdot e^{\beta \hat{r}_{i,t}(c)} \\
&= \sum_{i=1}^{K} \frac{p_{i,t}(c) - \alpha}{1 - K\alpha} e^{\beta \hat{r}_{i,t}(c)} \\
&\leq \frac{1}{1 - K\alpha} \sum_{i=1}^{K} (p_{i,t}(c) - \alpha) \left(1 + \beta \hat{r}_{i,t}(c) + (e - 2)\beta^2 \hat{r}_{i,t}(c)^2\right) \\
&\leq 1 + \frac{\beta}{1 - K\alpha} \sum_{i=1}^{K} p_{i,t}(c) \hat{r}_{i,t}(c) + \frac{(e-2)\beta^2}{1 - K\alpha} \sum_{i=1}^{K} p_{i,t}(c) \hat{r}_{i,t}(c)^2,
\end{aligned}
$$

where the first equation holds because for any $c \in [C]$, $w_{i,t+1}(c) = w_{i,t}(c) \cdot e^{\beta \hat{r}_{i,t}(c)}$, and the second equation follows because $p_{i,t}(c) = (1 - K\alpha)\frac{w_{i,t}(c)}{W_t(c)} + \alpha$.

In the first inequality, we have used the fact that $\beta \hat{r}_{i,t}(c) \leq \beta r_{i,t}(c)/\alpha \leq 1$ (since $\beta/\alpha \leq 1$ for any choice of $T$ and $K$), that $e^x \leq 1 + x + (e-2)x^2$ for $x \in [0,1]$, and that all rewards $r_{i,t}(c)$ are bounded in $[0,1]$. Now, using the fact that $\log(1 + x) \leq x$, we have that:

$$\log \frac{W_{t+1}(c)}{W_t(c)} \leq \frac{\beta}{1 - K\alpha} \sum_{i=1}^{K} p_{i,t}(c) \hat{r}_{i,t}(c) + \frac{(e-2)\beta^2}{1 - K\alpha} \sum_{i=1}^{K} p_{i,t}(c) \hat{r}_{i,t}(c)^2,$$

and therefore (summing over all $t$)

$$\log \frac{W_T(c)}{W_0(c)} \leq \frac{\beta}{1 - K\alpha} \sum_{t=1}^{T} \sum_{i=1}^{K} p_{i,t}(c) \hat{r}_{i,t}(c) + \frac{(e-2)\beta^2}{1 - K\alpha} \sum_{t=1}^{T} \sum_{i=1}^{K} p_{i,t}(c) \hat{r}_{i,t}(c)^2. \tag{12}$$

Recall that we compute regret against the optimal stationary policy $\pi(c) = \arg\max_i \sum_{t=1}^{T} r_{i,t}(c)$. Then,

$$
\begin{aligned}
\log \frac{W_T(c)}{W_0(c)} &\geq \log \frac{w_{\pi(c),T}(c)}{K} \\
&= \beta \sum_{t=1}^{T} \hat{r}_{\pi(c),t}(c) - \log K, \tag{13}
\end{aligned}
$$

where the first inequality holds because (i) $w_{i,0}(c) = 1$ for any $i \in [K]$ and as a result, $W_0(c) = K$, and (ii) $W_T(c) = \sum_{i=1}^{K} w_{i,T}(c) \geq w_{\pi(c),T}(c)$. From (12) and (13), we get

$$\frac{\beta}{1 - K\alpha} \sum_{t=1}^{T} \sum_{i=1}^{K} p_{i,t}(c)\hat{r}_{i,t}(c) + \frac{(e-2)\beta^2}{1 - K\alpha} \sum_{t=1}^{T} \sum_{i=1}^{K} p_{i,t}(c)\hat{r}_{i,t}(c)^2 \geq \beta \sum_{t=1}^{T} \hat{r}_{\pi(c),t}(c) - \log K. \quad (14)$$

Simplifying (14) (multiplying through by $(1 - K\alpha)/\beta^3$ and applying the fact that $r_{i,t}(c)$ is bounded), this becomes

$$\sum_{t=1}^{T} \hat{r}_{\pi(c),t}(c) - \sum_{t=1}^{T} \sum_{i=1}^{K} p_{i,t}(c)\hat{r}_{i,t}(c) \leq \frac{\log K}{\beta} + (e-2)\beta \sum_{t=1}^{T} \sum_{i=1}^{K} p_{i,t}(c)\hat{r}_{i,t}(c)^2 + KT\alpha. \quad (15)$$

We now take expectations (with respect to all randomness, both of the algorithm and of the contexts) of both sides of (14) and apply our bound (11) on the variance of $\hat{r}_{i,t}(c)$.

$$
\begin{aligned}
\sum_{t=1}^{T} r_{\pi(c),t}(c) - \sum_{t=1}^{T} \sum_{i=1}^{K} \mathbb{E}[p_{i,t}(c)]r_{i,t}(c) &\leq \frac{\log K}{\beta} + (e-2)\beta \sum_{t=1}^{T} \sum_{i=1}^{K} \frac{\mathbb{E}[p_{i,t}(c)]}{\alpha} r_{i,t}(c)^2 + KT\alpha \\
&\leq \frac{\log K}{\beta} + (e-2)\frac{\beta T}{\alpha} + KT\alpha \\
&\leq O(K^{1/3}T^{2/3}(\log K)^{1/3}) \quad (16)
\end{aligned}
$$

where this last inequality follows from the definition of $\alpha$ and $\beta$.

Now, note that the expected regret $\mathbb{E}[\text{Reg}(\mathcal{A})]$ of our algorithm is equal to

$$
\begin{aligned}
\mathbb{E}[\text{Reg}(\mathcal{A})] &= \mathbb{E}\left[ \sum_{t=1}^{T} r_{\pi(c_t),t}(c_t) - \sum_{t=1}^{T} r_{I_t(c_t),t}(c_t) \right] \\
&= \sum_{t=1}^{T} \mathbb{E}\left[ r_{\pi(c_t),t}(c_t) - r_{I_t(c_t),t}(c_t) \right] \\
&= \sum_{t=1}^{T} \sum_{c=1}^{C} \Pr[c] \, \mathbb{E}\left[ r_{\pi(c),t}(c) - r_{I_t(c),t}(c) \right] \\
&= \sum_{t=1}^{T} \sum_{c=1}^{C} \Pr[c] \left( r_{\pi(c),t}(c) - \mathbb{E}\left[ r_{I_t(c),t}(c) \right] \right)
\end{aligned}
$$

Considering the fact arm that $I_t$ is drawn from distribution $p_t(c)$, we get

$$
\begin{aligned}
\mathbb{E}[\text{Reg}(\mathcal{A})] &= \sum_{t=1}^{T} \sum_{c=1}^{C} \Pr[c] \left( r_{\pi(c),t}(c) - \sum_{i=1}^{K} \mathbb{E}[p_{i,t}(c)]r_{i,t}(c) \right) \\
&= \sum_{c=1}^{C} \Pr[c] \left( \sum_{t=1}^{T} r_{\pi(c),t}(c) - \sum_{t=1}^{T} \sum_{i=1}^{K} \mathbb{E}[p_{i,t}(c)]r_{i,t}(c) \right) \\
&\leq \sum_{c=1}^{C} \Pr[c] \cdot O(K^{1/3}T^{2/3}(\log K)^{1/3}) \\
&= O(K^{1/3}T^{2/3}(\log K)^{1/3}),
\end{aligned}
$$

where the inequality follows from (16).

$\square$

## A.3 Lower bound for learning to bid

In this section, will show that any algorithm for learning to bid in a first-price auction must incur at least $\Omega(T^{2/3})$ regret even if there is only one value (so no potential for cross-learning between contexts). To show this, we will use a reduction to the problem of dynamic pricing.

The problem of dynamic pricing is as follows. You must repeatedly (for $T$ rounds) sell an item to a buyer with value $x_t$ drawn iid from some unknown distribution $\mathcal{D}$. You do this by proposing a price $p_t$. If $x_t \geq p_t$, the buyer buys the item and you receive payment $p_t$ (alternatively, regret $(x_t - p_t)$); otherwise if $x_t < p_t$ the buyer does not buy the item and you receive regret $x_t$. The goal of this game is to maximize total revenue, or equivalently, minimize the total regret (with respect to the optimal fixed price $p^*$).

Kleinberg and Leighton [15] prove the following bounds on this problem.

**Theorem 24** (Theorem 4.3 in [15]). *For any $T$, there exists a family of distributions $\mathcal{P} = \{\mathcal{D}_i\}$ on $[0,1]$ such that if $\mathcal{D}$ is sampled uniformly from $\mathcal{P}$ and the buyer's valuations are sampled iid according to $\mathcal{D}$, any pricing strategy must incur expected regret $\Omega(T^{2/3})$.*

This lower bound can be matched (up to log factors) by discretizing (to $K = O(T^{1/3})$ intervals) and running EXP3.

We now show this lower bound immediately implies a lower bound on the learning to bid problem, even when there is only one context.

**Theorem 25.** *Any algorithm must incur $\Omega(T^{2/3})$ regret for the learning to bid in first price auctions problem, even if the value of the bidder is fixed (i.e. there is only one context).*

*Proof.* We will show how to use a learning algorithm for the learning to bid problem to solve the dynamic pricing problem.

Consider an instance of the learning to bid problem where $v_t = 1$ always (i.e. $\mathcal{D}_v$ is the singleton distribution supported on 1). If the bidder bids $b_t$ in this auction, then with probability $\Pr_{h \sim \mathcal{D}_h}[b_t \geq h]$ the bidder wins the auction and receives reward $(1 - b_t)$, and with probability $1 - P_t$ the bidder loses the auction and receives reward 0.

Now consider pricing when the value of the buyer is drawn from $\mathcal{D} = 1 - \mathcal{D}_h$ (that is, one can sample from $\mathcal{D}$ by sampling $x$ from $\mathcal{D}_h$ and returning $1 - x$). If set a price $p_t$ in this auction, then with probability $\Pr_{x \sim \mathcal{D}}[x \geq p_t]$, the item is sold and the seller receives reward $p_t$, and with probability $1 - \Pr_{x \sim \mathcal{D}}[x \geq p_t]$, the item is not sold and the seller receives reward 0.

But note that $\Pr_{x \sim \mathcal{D}}[x \geq p_t] = \Pr_{h \sim \mathcal{D}_h}[1 - h \geq p_t] = \Pr_{h \sim \mathcal{D}_h}[1 - p_t \geq h]$. In particular, setting a price of $p_t$ in the pricing problem with distribution $1 - \mathcal{D}_h$ results in the exact same feedback and rewards as bidding $1 - p_t$ in the learning to bid problem with distribution $\mathcal{D}_h$. One can therefore use any algorithm for the learning to bid problem to solve the dynamic pricing problem with the same regret guarantee; since Theorem 24 implies any learning algorithm must incur $\Omega(T^{2/3})$ regret on the dynamic pricing problem, it follows that any learning algorithm must incur $\Omega(T^{2/3})$ regret for the learning to bid problem. $\square$

## A.4 Omitted proofs

### A.4.1 Proof of Lemma 3

*Proof of Lemma 3.* Essentially, we must show that after observing arm $i$ $m_i(c)$ times, we no longer lose substantial regret from that arm in context $c$. Begin by noting that

$$
\sum_{i=1}^{K} \sum_{c=1}^{C} \sum_{t=1}^{T} \Delta_i(c) \mathbb{1}(I_t = i, c_t = c, \tau_{i,t} > m_i(c)) \leq \sum_{i=1}^{K} \sum_{c=1}^{C} \sum_{t=1}^{T} \mathbb{1}(I_t = i, c_t = c, \tau_{i,t} > m_i(c))
$$

$$
= \sum_{i=1}^{K} \sum_{t=1}^{T} \mathbb{1}(I_t = i, \tau_{i,t} > m_i(c_t)),
$$

896 where the inequality holds the reward of each arm $i$ and consequently gap $\Delta_i(c)$ is bounded in $[0, 1]$.

897 In expectation, this is equal to

$$\sum_{i=1}^{K}\sum_{t=1}^{T}\Pr[I_t = i, \tau_{i,t} > m_i(c_t)].$$

898 Now, define $U_{i,t}(c) = \bar{r}_{i,t}(c) + \omega(\tau_{i,t})$ to be the upper confidence bound for arm $i$ under context $c$ in
899 round $t$. Note that if $I_t = i$, then $U_{i,t-1}(c_t) \geq U_{j,t-1}(c_t)$ for any other arm $j$. This holds because the
900 algorithm chooses the arm with the highest upper confidence bound. It follows that (fixing $i$ and $t$)

$$\Pr[I_t = i, \tau_{i,t} > m_i(c_t)] \leq \Pr\left[U_{i,t-1}(c_t) \geq U_{i^*(c_t),t-1}(c_t), \tau_{i,t} > m_i(c_t)\right].$$

901 Define $t_i(n)$ to be the minimum round $t$ such that $\tau_{i,t} = n$, and define $\bar{x}_{i,n}(c) = \bar{r}_{i,t_i(n)}(c)$ (in other
902 words, $\bar{x}_{i,n}(c)$ is the average value of the first $n$ rewards from arm $i$, in context $c$). Note that if
903 $\tau_{i,t} \geq m_i(c)$, then $U_{i,t-1}(c) \geq U_{i^*(c),t-1}(c)$ implies that

$$\max_{m_i(c_t) \leq n \leq t} \bar{x}_{i,n}(c) + \omega(n) \geq \min_{0 < n' < t} \bar{x}_{i^*(c),n'}(c) + \omega(n').$$

904 We can therefore write

$$\Pr\left[U_{i,t-1}(c_t) \geq U_{i^*(c_t),t-1}(c_t), \tau_{i,t} > m_i(c_t)\right]$$

$$\leq \quad \Pr\left[\max_{m_i(c_t) \leq n \leq t} \bar{x}_{i,n}(c_t) + \omega(n) \geq \min_{0 < n' < t} \bar{x}_{i^*(c_t),n'}(c_t) + \omega(n')\right]$$

$$\leq \quad \sum_{n=m_i(c_t)}^{t}\sum_{n'=1}^{t} \Pr\left[\bar{x}_{i,n}(c_t) + \omega(n) \geq \bar{x}_{i^*(c_t),n'}(c_t) + \omega(n')\right].$$

905 Finally, observe that if $\bar{x}_{i,n}(c_t) + \omega(n) \geq \bar{x}_{i^*(c_t),n'}(c_t) + \omega(n')$, then one of the following events
906 must occur:

907     1. $\bar{x}_{i^*(c_t),n'}(c_t) \leq \mu^*(c_t) - \omega(n')$.

908     2. $\bar{x}_{i,n}(c_t) \geq \mu_i(c_t) + \omega(n)$.

909     3. $\mu^*(c_t) < \mu_i(c_t) + 2\omega(n)$.

910 Now, recall that $m_i(c) = \frac{8\log T}{\Delta_i(c)^2}$. Note that since $n \geq m_i(c_t)$, we have that $\omega(n) \leq \omega(m_i(c_t)) \leq$
911 $\Delta_i(c_t)/2$, so $\mu_i(c_t) + 2\omega(n) \leq \mu_i(c_t) + \Delta_i(c_t) \leq \mu^*(c_t)$, and therefore the third event can never
912 occur. Since the first two events both occur with probability at most $t^{-4}$ (by Hoeffding's inequality),
913 we have that

$$\Pr[I_t = i, \tau_{i,t} > m_i(c_t)] \quad \leq \quad \sum_{n=m_i(c_t)}^{t}\sum_{n'=1}^{t} \Pr\left[\bar{x}_{i,n}(c_t) + \omega(n) \geq \bar{x}_{i^*(c_t),n'}(c_t) + \omega(n')\right]$$

$$\leq \quad \sum_{n=m_i(c_t)}^{t}\sum_{n'=1}^{t} 2t^{-4} \leq 2t^{-2}.$$

914 Further summing this over all $i \in [K]$ and $t \in [T]$, we have that

$$\sum_{i=1}^{K}\sum_{t=1}^{T}\Pr[I_t = i, \tau_{i,t} > m_i(c_t)] \leq \frac{K\pi^2}{3},$$

915                                                                              $\square$

 **A.4.2   Proof of Theorem 5**

917 *Proof of Theorem 5.* The proof is similar to that of Theorem 23. Begin by defining the estimator

$$\hat{r}_{i,t}(c) = \frac{r_{i,t}(c)}{\sum_{c'} \Pr[c'] \cdot p_{i,t}(c')} \cdot \mathbb{1}(I_t = i).$$

918 Note that

$$\Pr[I_t = i] = \sum_{c'} \Pr[c'] \cdot p_{i,t}(c'),$$

919 so taking expectations over the algorithm's choice of $I_t$, we have that

$$\mathbb{E}[\hat{r}_{i,t}(c)] = r_{i,t}(c),$$

920 and

$$\mathbb{E}[\hat{r}_{i,t}(c)^2] = \frac{r_{i,t}(c)^2}{\sum_{c'} \Pr[c'] \cdot p_{i,t}(c')}.$$

921 Define $W_t(c) = \sum_{i=1}^{K} w_{i,t}(c)$. Now, proceeding in the same way as the proof of Theorem 23, we
922 arrive at the inequality

$$\sum_{t=1}^{T} \hat{r}_{\pi(c),t}(c) - \sum_{t=1}^{T} \sum_{i=1}^{K} p_{i,t}(c)\hat{r}_{i,t}(c) \le \frac{\log K}{\beta} + (e-2)\beta \sum_{t=1}^{T} \sum_{i=1}^{K} p_{i,t}(c)\hat{r}_{i,t}(c)^2 + KT\alpha. \quad (17)$$

923 We now take expectations (with respect to all randomness, both of the algorithm and of the contexts)
924 of both sides of (17).

$$\sum_{t=1}^{T} r_{\pi(c),t}(c) - \sum_{t=1}^{T} \sum_{i=1}^{K} \mathbb{E}[p_{i,t}(c)]r_{i,t}(c)$$

$$\le \frac{\log K}{\beta} + (e-2)\beta \sum_{t=1}^{T} \sum_{i=1}^{K} \mathbb{E}\left[ \frac{p_{i,t}(c)}{\sum_{c'} \Pr[c'] \cdot p_{i,t}(c')} \right] r_{i,t}(c)^2 + KT\alpha. \quad (18)$$

925 Note that the expected regret $\mathbb{E}[\text{Reg}(\mathcal{A})]$ of our algorithm is equal to

$$\mathbb{E}[\text{Reg}(\mathcal{A})] = \mathbb{E}\left[ \sum_{t=1}^{T} r_{\pi(c_t),t}(c_t) - \sum_{t=1}^{T} r_{I_t(c_t),t}(c_t) \right]$$

$$= \sum_{t=1}^{T} \mathbb{E}\left[ r_{\pi(c_t),t}(c_t) - r_{I_t(c_t),t}(c_t) \right]$$

$$= \sum_{t=1}^{T} \sum_{c=1}^{C} \Pr[c]\, \mathbb{E}\left[ r_{\pi(c),t}(c) - r_{I_t(c),t}(c) \right]$$

$$= \sum_{t=1}^{T} \sum_{c=1}^{C} \Pr[c] \left( r_{\pi(c),t}(c) - \mathbb{E}\left[ r_{I_t(c),t}(c) \right] \right)$$

926 Since arm $I_t$ is drawn from distribution $p_t(c)$, we have

$$\mathbb{E}[\text{Reg}(\mathcal{A})] = \sum_{t=1}^{T}\sum_{c=1}^{C}\Pr[c]\left(r_{\pi(c),t}(c) - \sum_{i=1}^{K}\mathbb{E}[p_{i,t}(c)]r_{i,t}(c)\right)$$

$$= \sum_{c}\Pr[c]\left(\sum_{t=1}^{T}r_{\pi(c),t}(c) - \sum_{t=1}^{T}\sum_{i=1}^{K}\mathbb{E}[p_{i,t}(c)]r_{i,t}(c)\right)$$

927 From (18), we get that

$$\mathbb{E}[\text{Reg}(\mathcal{A})] \leq \sum_{c=1}^{C}\Pr[c]\left(\frac{\log K}{\beta} + (e-2)\beta\sum_{t=1}^{T}\sum_{i=1}^{K}\mathbb{E}\left[\frac{p_{i,t}(c)}{\sum_{c'}\Pr[c']\cdot p_{i,t}(c')}\right]r_{i,t}(c)^2 + KT\alpha\right)$$

$$= \frac{\log K}{\beta} + (e-2)\beta\sum_{t=1}^{T}\sum_{i=1}^{K}\sum_{c=1}^{C}\Pr[c]\cdot\mathbb{E}\left[\frac{p_{i,t}(c)}{\sum_{c'}\Pr[c']\cdot p_{i,t}(c')}\right]r_{i,t}(c)^2 + KT\alpha$$

$$\leq \frac{\log K}{\beta} + (e-2)\beta KT + KT\alpha$$

$$= O(\sqrt{KT\log K}).$$

928 Here the final inequality holds since $r_{i,t}(c)$ is bounded in $[0,1]$. $\qquad\square$

### A.4.3 Proof of Lemma 19

930 *Proof of Lemma 19.* Consider the following distribution over instances of the multi-armed bandit
931 problem. Let $\varepsilon = \Theta(\sqrt{K/T})$ (the precise value to be chosen later). An $i$ is drawn uniformly at
932 random from $[K]$. The rewards from arm $i$ are distributed according to $B((1+\varepsilon)/2)$, and the arms
933 for all $j \neq i$ are distributed according to $B((1-\varepsilon)/2)$ (where here $B(p)$ is the Bernoulli distribution
934 with probability $p$).

935 We wish to claim that at any round $t \leq T$, the probability any learner plays the optimal arm $i$ is less
936 than $1/2$, and therefore the learner must incur $\Omega(\varepsilon) = \Omega(\sqrt{K/T})$ regret this round. This is therefore
937 a best-arm identification problem. Theorem 4 in [3] implies there exists some $\varepsilon = \Theta(\sqrt{K/T})$ such
938 that this result holds for our distribution of instances. $\qquad\square$