[Reviews · NeurIPS 2019]

Reviewer 1



This paper studies the contextual bandits with cross learning. Existing literature which does not assume any structure among the contexts establishes a weak O(\sqrt{CKT}) bound. In this paper, the authors assume that the rewards corresponding to every context is observable and prove a O(\sqrt{KT}) bound under two settings; The setting where the contexts are adversarial, but the rewards are stochastic and the setting where the contexts are stochastic, but the rewards are adversarial. The paper also proves a lower bound of O(\sqrt{CKT}) for the setting when the rewards and contexts are both adversarial arguing that cross learning does not help when rewards and contexts are both adversarially generated. The paper abstracts a number of interesting applications into a new theoretical problem and presents two different algorithms, UCB1.CL and EXP3.CL along with their theoretical analysis. The paper acknowledges that UCB.1CL and EXP3.CL are respectively minor variations of existing bandit algorithms, UCB-1 and EXP3 and claims that their analysis requires significantly new ideas. However, I think the paper fails to convincingly argue that existing proof techniques (for both UCB-1 and EXP3) do not extend to the problem setting under consideration. The paper fails to argue why the “optimism under uncertainty” proof argument (for example see Section 4 in [1]) do not carry over for proving the regret bound of UCB.1CL. Since we observe the rewards corresponding to all the contexts for any selected arm and hence can construct the corresponding confidence intervals, the preceding argument can be used verbatim to prove the desired bound. [EDIT] After considering the author response, this reviewer no longer holds the earlier concern about the correctness of the lower bound. The reviewer has changed their score accordingly. References [1] Agrawal, Shipra, and Nikhil R. Devanur. "Bandits with concave rewards and convex knapsacks." Proceedings of the fifteenth ACM conference on Economics and computation. ACM, 2014.

Reviewer 2



When contexts are adversarial, a UCB-style algorithm that operates pointwise achieves good regret, especially in the partial cross-learning case where the average clique size in the reward elicitation graph is the relevant quantity. This algorithm is computationally viable when rewards have a convenient parametric form eliding the need to enumerate contexts. This algorithm seems likely to find practical application. The EXP3 variant is more of theoretical interest and is unlikely to be practically applied without additional research, e.g., eliminating the need to know the context distribution.

Reviewer 3



This paper explores contextual bandits with cross learning, i.e., when an arm i is pulled with context c, the learner not only observes the rewards for arm i in context c, but also observes the reward of arm i for all other contexts. The authors investigate the settings in which the context/rewards are stochastic or adversarial and show the corresponding regret bounds (or regret lower bound). Overall I think it’s a well-written paper. The results seem sound and reasonable (though I didn’t carefully check the proof), and in particular, I found the analysis of UCB1.CL (which drops the dependency on C in the regret bounds) to be non-trivial and interesting. I would lean towards acceptance. Other comments: I’m not entirely convinced that auctions are best formulated as contextual bandits with cross-learning (which seems to be the major motivating example of this paper). As the authors also pointed out in the experiments, it requires the assumption that other bidders’ valuations to be independent of the bidder’s valuation. The other examples illustrated in the paper (e.g., multi-armed bandits with exogenous costs) are reasonable though don’t seem as impactful/pressing as the auction settings. Honestly, though I don’t have other applications in mind, I think the setting is general enough (especially with the added discussion on partial cross-learning) and worth investigating. However, given one of the main contributions of the paper is to introduce this cross learning setting, it would be nice if the papers can provide stronger motivations for this setting. In the experiment setup, the authors assume the “true” valuations of some participant is known. How is this achieved? Are there additional assumptions on bidders’ beliefs, so you can infer the bidders’ values from bids in first price auctions?

[Author Response · NeurIPS 2019]

We thank all the reviewers for their detailed feedback.

**Reviewer 3**:

We believe the proof of our lower bound is correct as written. In our lower bound, the rewards $r_{i,t}(c_j)$ do have an
explicit dependence on $c_j$; specifically we write that "assign rewards so that $r_{i,t}(c) = 0$ if $t$ is in the $j$th epoch and
$c \neq c_j$" (otherwise we assign rewards according to some hard bandit instance). The dependence on $c_j$ comes from the
$c \neq c_j$ clause. In other words, during the $j$th epoch in which the context is $c_j$, the rewards of all other contexts are zero.

We do not understand what the reviewer means by "extending the existing proof of EXP3" to the contextual cross-
learning setting. We certainly use the general proof structure of EXP3 (as do very many learning algorithms in settings
with adversarial rewards), but the core technical contribution here is figuring out how to extend the EXP3 algorithm to
the contextual setting with cross-learning. The most direct such approach (updating all arms/contexts you have access
to the exact same way as in EXP3) results in our EXP3.CL-U algorithm, which results in a suboptimal $\tilde{O}(K^{1/3}T^{2/3})$
regret by the standard EXP3 analysis. Extending this to EXP3.CL involves constructing a low-variance unbiased
estimator via importance sampling, which we believe is a novel contribution in this setting.

We believe that the proof argument in Section 4 of the "Bandits with concave rewards and concave knapsacks" paper
can be adapted to the contextual setting to recover an $\tilde{O}(\sqrt{KT})$ regret bound for UCB1.CL (we thank the reviewer
for pointing out this alternate proof). However, to the best of our knowledge, this technique 1. does not allow us to
prove gap-dependent bounds that have logarithmic dependency in $T$ and 2. does not extend to the partial cross-learning
setting (here the inequality in (2) breaks). Since our technique in the proof of Theorem 1 nicely generalizes to these
settings, we believe it has technical merit. We plan to mention this in the updated version of the paper.

**Reviewer 4**:

We thank the reviewer for suggesting an experimental comparison with LinUCB; we agree it is an interesting direction
for future work.

**Reviewer 5**:

We understand that the assumption that the learner's value is independent from the other bidder's bids is a strong
assumption. We have the following thoughts about this subject:

- If we condition on enough public features of the query, this can significantly decrease the correlation between
different bids (of course, this leads to an interesting question of how to cross-learn between these new contexts
given by public features).

- In particular, many advertisers in display advertising markets base bids on cookies, which are information
stored on users' browsers. Because cookies are private, cookie-based bids are typically weakly correlated. For
example, Amazon knows that a user is interested in purchasing shoes because she searched for shoes on its
website, but a competitor such as Macy's might not have this information.

- In our experiments, the data does not have the property that the values (contexts) are independent of the other
bids (in fact, the linear correlation between the two is sizeable). Nonetheless, the cross-learning algorithms
designed in the paper seem to perform well, and understanding this robustness is an interesting direction for
future work.

- Part of the motivation for the development of contextual bandits with partial cross-learning is exactly to
address this issue; even if correlation is present, you might have some idea of which other contexts it is safe to
cross-learn over.

The reviewer asked where we obtain the values for the bidder in our simulation. This bidder directly provided their true
valuation for each query (specifically, what they would have bid for the same query in a truthful second price-auction)
to us for the purpose of this research project. We will clarify this in the updated version of the paper.

[Meta-Review · NeurIPS 2019]

Reviewers expressed some concerns in their initial reviews, but are satisfied with the author response. That said, the authors are strongly encouraged to improve the motivation, particularly to discuss in more detail how their motivating examples correspond to the assumptions in the model.